# Erosion after an extreme storm event in an arid fluvial system of the southern Atacama Desert: an assessment of magnitude, return time, and conditioning factors of erosion and debris flows generation

Germán Aguilar[1], Albert Cabré[1,2], Víctor Fredes[1,3,4], and Bruno Villela[1,3]

[1]Advanced Mining Technology Center, Facultad de Ciencias Físicas y Matemáticas, Universidad de Chile, Avenida Tupper 2007, Santiago, Chile
[2]Departamento de Ciencias Geológicas, Universidad Católica del Norte, Avenida Angamos 0610, Antofagasta, Chile
[3]Departamento de Geología, Facultad de Ciencias Físicas y Matemáticas, Universidad de Chile, Plaza Ercilla 803, Santiago, Chile
[4]BGC Ingeniería Ltda., Apoquindo 3039, Santiago, Chile

**Correspondence:** Germán Aguilar (german.aguilar@amtc.cl)

**Abstract.** The contribution of an individual extreme storm event to long-term erosion rates has been estimated for the first time in the Atacama Desert. A mean erosion of 1.3 mm has been calculated for the March 2015 event that impacted the southernmost Atacama Desert. The estimated erosion is consistent with millennial erosion rates and the previously reported return times of high sediment discharge events in the study area. This is significant because erosion rates, related to events of high sediment discharge in arid fluvial systems, are difficult to measure with sediment loading due to destruction of gauges by devastating flashfloods and therefore have not been directly measured yet. During the March 2015 storm, debris flows were reported as the main sediment transport process, while gullies and channels erosion were the main source of sediments that generated debris flows reaching the tributary junctions and the trunk valleys. Sediment yield to tributary outlets is highly dependent on the ability of catchments to store sediments in stream networks between storms. The largest tributary catchments, the high hydrological hierarchy, the low topographic gradient and the gentle slopes are the most determinant factors to generate debris flows capable of reaching alluvial fans in any storm event from large sediment volumes stored in the stream networks. Our findings better assess the susceptibility to debris flow of arid catchments, which is significant for the southernmost valleys of the Atacama Desert because human settlements and industries are mostly established in alluvial fans.

## 1 Introduction

The hydrology of the Atacama Desert is characterized by rivers that flow from the high Andean mountain ranges fed from snow, glaciers and permafrost melting (Favier et al., 2009; Gascoin et al., 2011). In this scenario, the presence of perennial

rivers is restricted to the trunk valleys, while the tributary valleys of the mid-mountainous region and of the Andean foothills host ephemeral stream networks. High runoff and sediment discharges in these ephemeral stream networks are triggered by intense rare rainfall events. The influence of extreme storms has resulted in significant erosion runoff events in the past recorded by coarse grain size layers in lacustrine and marine archives per the past 5,500 years in the Atacama Desert (Ortlieb, 1994, 1995; Veit, 1996; Rein et al., 2005; Maldonado and Villagrán, 2006; Vargas et al., 2006; Martel-Cea et al., 2016; Tiner et al., 2018; Ortega et al., 2012, 2019). The return time of extreme runoff erosive events decreased from 1 event/210 yrs. to 1 event/ 40 yrs. during the last 1000 yr. This is interpreted as an intensification of ENSO storms activity in Northern Chile (Ortega et al. (2019) and references therein).

The Atacama Desert has historically been impacted by extreme storms that generally occur during southern hemisphere winters. Almost all records available are from local newspapers and have recently been compiled in Ortega et al. (2019); Vargas et al. (2018). Most of the events triggered flash floods and debris flows that affected preferred sites for human settlements, such as fluvial plains and alluvial fans. In March 2015, a storm impacted a large area of the Atacama Desert (Barret et al., 2016; Wilcox et al., 2016; Bozkurt et al., 2016; Jordan et al., 2019). This event occurred during the southern hemisphere summer in an area where the highest mountain ranges are above 3500 m a.s.l. and generally experience precipitation in the form of snow. The high elevation of the zero isotherm during the March 2015 event caused erosion in areas where snow usually delays run-off (Wilcox et al., 2016; Jordan et al., 2019). This catastrophic hydrometeorological event caused human casualties and considerable economic losses in the main cities located at El Salado River, El Copiapó River, El Huasco River and El Elqui River.

Extreme storm events contribute greatly to erosion in arid zones (Tarr, 1890; Coppus and Imeson, 2002). Their influence has been highlighted in Carretier et al. (2018) in the Andean catchments of Chile. This has been reinforced by the differences in the concentration of Terrestrial Cosmogenic Nuclides in sediments supplied by debris flows against sediments supplied by the combination of fluvial processes in the El Huasco river valley (Aguilar et al., 2014) and, for other arid fluvial systems in Chile (Carretier et al., 2015). Therefore, the magnitude and distribution of erosion after extreme storms is of great interest for both modeling landscape evolution and for analyzing sediment cascades dynamics in the southernmost Atacama Desert. However, the precise quantification of catchment erosion during a major flood event has not yet been established. This might be explained because of the lack of high-resolution digital elevation models (e.g. Lidar point clouds in Anderson et al. (2015)) and due to the absence of detailed topography available for the Atacama Desert. The problem is compounded by the lack or scarcity of meteorological and fluviometric stations to measure the intensity of the hourly rain and the sediment load yielding, respectively.

The aim of this study is to evaluate erosion during the March 2015 storm in catchments of El Huasco river valley (Fig. 1). We have calculated the erosion within the tributary catchments (total area of 1,500 km$^2$) based on volumes of debris flow deposits measured in tributary junction alluvial fans after the storm. The calculated erosion agrees with the Holocene erosion rates presented in Aguilar et al. (2014) by the concentrations of Terrestrial Cosmogenic Nuclides in stream sediments. Therefore, we illustrate how erosion in this arid area is dictated by the repetition of individual extreme storm events such as the March 2015 event. In addition, the statistical evaluation proves that the topographic attributes of the catchments facilitate the storage

of debris within them, which is ultimately significant for the generation of debris flow. These results suggest that debris-storage should be considered to adequately define the susceptibility to the generation of debris flows within the catchments, and these are significant for the assessment of hydrometeorological hazards.

## 2   Study area

El Huasco River valley (29°S) is one of the main fluvial systems in the climatic transition area between the hyper-arid core of the Atacama Desert and the semiarid region of Central Chile. This river has an hydrological basin of 9,800 km$^2$ from the continental water division in the East to the Pacific Ocean in the West (Fig. 1). Three main physiographic units are present at this latitude for the western slope of the Andes: the Coastal Cordillera (CC), which is located at the lowest elevations in the western area of the El Huasco River watershed, the Principal Cordillera (PC) in the middle elevations and the Frontal Cordillera

(FC) in the high mountain ranges to the East. The Huasco River takes its name from the union of two main rivers on the border between the Coastal and the Principal Cordilleras: the El Carmen and the El Tránsito Rivers. These two rivers have permanent run-off from snow and permafrost melt from the highest tributaries located at the Frontal Cordillera with summits rising 6,000 m a.s.l. Glaciers and permafrost have been reported in the Frontal Cordillera above  4,000 m a.s.l. and largest record of glacial advance in the Late Pleistocene does not go beyond 3,200 m a.s.l. (Aguilar, 2010). Rainfall in the Principal Cordillera provides

additional runoff that generally occurs during the southern hemisphere winter.

The study area (1,500 km$^2$) in which this paper focuses is situated in the Principal Cordillera. In this physiographic unit, the igneous-metamorphic Permo-Triassic and Mesozoic rocks exhumed during the Cenozoic represent the highest mountain ranges. These Paleozoic rocks are subdivided based on Salazar et al. (2013) into: the Pampa Gneisses, the Tránsito Metamorphic Complex and the Chanchoquín Plutonic Complex. Volcano-sedimentary lithological units from the Mesozoic are also present

between these Permo-Triassic tectonic-blocks. These are the San Félix Formation, the Lautaro Formation, and the Lagunillas Formation. These lithological units are intruded by several Cenozoic plutons and are covered in some areas by Cenozoic unconsolidated gravel deposits (e.g. the Atacama Gravels) and volcanic deposits (Salazar et al., 2013). Five geological groups were defined by Fredes (2016) considering a rock strength classification (González de Vallejo, 2002) of the geological units of Salazar et al. (2013) (Fig. 2). Geo1 corresponds to intrusive rocks, Geo2 corresponds to volcanic and coarse-sedimentary rocks,

Geo3 corresponds to metamorphic rocks, Geo4 corresponds to fine grained rocks (shales and siltstones) and Geo5 corresponds to unconsolidated Pleistocene-Holocene deposits.

The quantification and characterization of erosion and sedimentation in the El Huasco river valley has been interpreted by different authors as an induced climatic and tectonic signal (Aguilar et al., 2011, 2013, 2014; Rossel et al., 2018). On a 10$^6$ year time-scale, tectonic and climatic factors combining aridization and Andean uplift are coupled to develop a transient landscape

organization (Aguilar et al., 2013). The calculated mean erosion rates of 0.03-0.08 mm/yr during the last 10-6 Ma do not reach the total degradation of the Miocene surfaces located at the highest altitudes (Aguilar et al., 2011). Millennial erosion rates (10$^4$ year) measured by concentrations of Terrestrial Cosmogenic Nuclides (TCN) in river sediments are consistent with erosion rates of 10$^6$ years (Aguilar et al., 2014). The grain size dependent concentration of TCN in the alluvial stream sediments would

be induced by the contribution of gravels from the debris flows in the Holocene (Aguilar et al., 2014). Episodic erosion and sedimentation associated with debris flow deposits are recorded in the Holocene stratigraphic arrangements of the tributary-junction alluvial fans situated at the Principal Cordillera (Veit, 1996; Aguilar, 2010; Cabré et al., 2017, 2019).

## 3   Rainfall data and erosion processes operating during the March 2015 storm event

The Principal Cordillera is an area that is generally impacted by extreme storm events that trigger debris flows in the tributary catchments (Cabré et al., 2020). These storms are spatially heterogeneous and their influence on the landscape is evidenced by multiple rills and gullies on the hillslopes together with erosion of channels within drainage networks (Fig. 3 and Fig. 4). The March 2015 storm event lasted 4 days in the El Huasco River watershed (23-26th) although it was not until the 24th that locals reported debris flows (Cabré et al., 2020). Figure 1 presents the precipitation map linked to the storm event of March 2015 that affected the Huasco River valley. This precipitation map shows that the most impacted areas were the Principal Cordillera, and in particular the area studied with accumulated daily rainfall greater than 50 mm.

Hillslopes of tributary catchments of El Huasco river were impacted by a combination of water erosion processes during the March 23rd -26th 2015 storm. In which, amongst all, the erosion of the upper layer of mantled hillslopes was produced when the water was concentrated and formed rills or gullies (Fig. 3 and Fig. 4). Rills begin to form from the overland flow when a critical shear stress is reached (Horton, 1945). If overland flow meets bare hillslopes characteristically of arid zones, hillslope erosion occurs in any mean-average storm because of the high erodibility of the upper soil layer. On the other hand, narrow and deep channels (gullies) are relatively permanent on the hillslopes and usually allow the transmission of water and sediment from high areas of the hillslope to the main drainage network within the catchments.

The March 2015 storm event severely impacted the alluvial channels of the stream-network that store large volumes of sediment within the tributary catchments (Fig. 3b). Sediment entrainment resulted in different debris flows pulses with different sediment to water ratios. The debris flows were followed by a strong incision that facilitated the transport of sediment from alluviated channels downsystem. When these debris flows lost confinement at the outlet of the tributary catchments, their deposition occurred (Fig. 3fg). Table 1 presents the main characteristics of sedimentary facies of debris flows deposited during the storm in the tributary-junction alluvial fans (Cabré et al., 2020). For details on the spatial and temporal distribution of debris flows pulses in the fans, readers are referred to Cabré et al. (2020).

## 4   Methods

### 4.1   Debris flow deposits volumes and erosion measures

Immediately after the March 2015 storm, we conducted a field survey in El Huasco River valley. We have characterized the erosion processes within the catchments, both from fieldwork observations and from the analysis of optical satellite images retrieved from Planet Team (2017). Unfortunately, the available 3m/pixel images for Planet Scope Images and 6.5m/pixel for RapidEye Images do not help in identifying rills and small gullies. Therefore, field observations are crucial to identify these

erosion indicators. We observed debris flow deposition on the alluvial fan surfaces and the formation of alluvial fan at the confluence of the tributaries with the trunk valley. The analysis of the deposits showed different rheology of the debris flows ranging from cohesive debris flows and hyper-concentrated flows to mud flows. Here, to simplify statistical analysis we do not differentiate between flow-types and henceforth we will include all under the term "debris flow" sensu stricto. The debris flow deposits volumes were estimated using Equation 1. This equation calculates the volume of the fans assuming a simplistic cone geometry and is adapted from the Campbell and Church (2003) equation for the colluvial fan volume calculations.

$$Volume = (\frac{wlt}{2})\frac{\pi}{3} \tag{1}$$

The width (w) and thickness (t) of 15 debris flow deposits in the alluvial fans were measured in the field with a measuring tape, while their axial length (l) was measured in the available satellite images. The width and length of the 33 debris flows deposits on fans was measured from RapidEye images. Based on the fieldwork measurements, a thickness between 0.5 and 1.5 m was considered for the debris flows whose length and width were measured on RapidEye images. The volumes have been corrected with a bulking factor that considers a porosity value of 30% (Nicoletti and Sorriso-Valvo, 1991). The data used for the volume calculations of debris flow deposits and its accuracy is available in the supplementary data 1.

We have calculated the mean erosion (mean surface lowering in millimeters) after the March 2015 storm event for each of the tributary catchments and for all the catchments within the study area with the volumes of debris flow deposits (Fig. 1, tributaries inside dashed-line white box). The tributary catchments were extracted in a Geographic Information System environment. The Digital Elevation Model used was retrieved from NASA/JPL-Caltech with a nominal spatial pixel resolution of 30 x 30m (https://asterweb.jpl.nasa.gov/gdem.asp) and a nominal vertical resolution of 1.0 m. The data used for the erosion calculations is available in the supplementary data 2.

## 4.2 Topographic attributes of catchments

Topographic attributes of 124 catchments were selected in order to analyze their influence on debris flow generation and erosion processes after the March 2015 storm. In order to characterize the tributary catchments, the topographic attributes were selected based on their influence on different processes such as peak flow generation and debris storage (Strahler, 1958; Melton, 1965; Howard and Kerby, 1983; Wilford et al., 2004): Area, Length (straight-line between outlet and head-water of tributary catchments), Maximum elevation, Gradient (inclination of length), Average Slope, Gravelius index (Shape), Hypsometry, Melton ratio (index of roughness that normalizes relief by area; Melton (1957)), Drainage density, maximum Strahler order, concavity, and steepness index of the main thalweg. The data used is available in the supplementary data 2.

A statistical analysis was calculated in order to find outliers or anomalies in the topographic attributes that might control debris flow generation in tributary catchments. The significance of each attribute in debris flows generation is determined by an Analysis of variance (ANOVA). The ANOVA determines if the mean values are similar between the catchment classes that generated debris flows against catchments that did not generate debris flows. The proposed tests consisted in splitting the total data variance into several components (between groups and within groups) and in comparing these components with a Fisher

mean test with a critical value of 0.05 (Box et al., 1978; Davis, 2002). Principal Components Analysis (PCA) was performed to group the variables related to debris flow generation in common factors that might explain the variance of the catchments attributes, reducing the number of variables and providing a weighting factor for each attribute within the group (Levy and Varela, 2003). A normalized classification of catchment-clustering was performed by the weighted mean of the attributes that control the debris flow generation on each catchment.

## 4.3 Geological attributes of catchments

Geological groups were defined in order to analyze the influence of the rock units on the debris-flow generation based on the map of Salazar et al. (2013). Eight hundred and forty Schmidt Hammer (SH) rebound values were measured in little-weathered bedrock outcrops in 41 selected field stations to provide a rock strength classification of the geological groups as similarly presented in Stokes and Mather (2015) for tributary catchments in Morocco. We did not measure unconsolidated sediments because SH values are below instrumental detection (SH < 10) (Selby, 1993). The rock strength classification includes more than 100 measurements for each of the geological groups. Additionally, a statistical analysis was performed for each geological group. Finally, we calculated the normalized SH values for the catchments considering areal percentage of the geological groups, Mean Schmidt Hammer values (MeanSHn) and IQR Schmidt Hammer values (IQRSHn). The MeanSHn values represent a little-weathered-rock strength index for the catchments whilst IQRSHn values represent an index of the weathering and cracking status of the bedrock as suggested in Roda-Boluda et al. (2018). The data used is available in the supplementary data 2.

## 5 Results

### 5.1 Debris flows volumes and mean erosion

Field data used for the volume calculations of debris flow deposits and its accuracy is available in the supplementary data 1. Debris flows that reached the tributary junctions with the trunk valley and produced deposits greater than 500 m$^3$ during the March 2015 event were identified in 49 of the one 124 tributary junctions in the studied area. The remaining 75 catchments did not present debris flows deposits greater to 500 m$^3$ of sediments in the tributary junctions. The volumes of the debris flow deposits vary between 530 and 234,000 m$^3$. The propagation of errors entails an uncertainty of -19% and +22% for the volumes calculated with field measurements and -63% and +93% for volumes estimated by satellite image measurements. The total sediment yield by debris flows to the tributary junction alluvial fans is approximately 1 million m$^3$ (-34%/+46%). The mean erosion values (mean surface lowering) for the tributary catchments was obtained from the calculated volumes of debris flow deposits that reached the alluvial fans. These erosion values vary between 0.3 and 35 mm (Fig. 5). If we consider the total area of the tributary catchments (including the catchments where debris flow were absent in the tributary junctions), the total mean erosion reported for the extreme storm event of March 2015 is 1.3 mm for an area of 1,500 km$^2$.

## 5.2 Catchment topographic attributes

Topographic attribute values for 124 tributary catchments and the statistical analysis are available in supplementary data 2. Six topographic attributes (Area, Length, Strahler Order, Slope, Melton ratio and Relief ratio) were proved to be significant in the generation of debris flow, with a level of significance below 0.05 in the ANOVA. Two groups resulted from the PCA analysis and each includes three topographic attributes. Group 1 integrates the size of the catchment and the hydrological hierarchy of the drainage network that includes: Area (A), Length (L) and Strahler order (O). Group 2 integrates the catchment relief and includes average Slope (S), Gradient (G) and Melton ratio (M). From these two groups, two conditioning-factors were calculated: Size Factor and Relief Factor. These factors correspond to a linear combination of the resulting PCA attributes and the weight of each attribute is set by its weight. Therefore, two conditioning-factor values resulted from the addition of three normalized attribute values (Xn) weighted by the weighting factor calculated by the PCA.

$$SizeFactor = (An * 0.92) + (Ln * 0.90) + (On * 0.89) \tag{2}$$

$$ReliefFactor = (Sn * 0.95) + (Gn * 0.80) + (Mn * 0.70) \tag{3}$$

Size Factor and Relief Factor show an inverse correlation for catchments that generated debris flows (49 catchments) and for catchments where clean water flows flushed downsystem in the tributary junctions (75 catchments without debris flows generation) (Fig. 6a). The inverse correlation is also observed in the percentage of catchments with different factor ranges, because the percentage of catchments that generated debris flows in the tributary junctions increases with Size Factor while it decreases with Relief Factor (Fig. 7ab). In fact, debris flows were present only in 18% of very small and low hierarchical catchments (low values of Size Factor, 0.05-0.25). In contrast, debris flows were present in 57% of large and high hierarchical catchments (high values of Size Factor, 0.75-1.50). Debris flows occurred only in 9% of the catchments with high mean slope values and high topographic gradient (high-Relief Factor, 1.75-2.00) while in 50% of the catchments with low slope values and low topographic gradients, debris flow occurred (low-Relief Factor, 0.25-1.25). These results suggest that debris flows were generated in the tributary junctions from larger and more hierarchical catchments with low topographic gradients and lower mean slopes, whereas sparsely generated from smaller and steeper catchments (Table 2).

The six topographic attributes (Area, Length, Strahler Order, Average Slope, Gradient, and Melton ratio) considered in the two conditioning-factors are reclassified to 0, 1 and 2 according to the percentage of catchments favorable to the generation of debris flow. Finally, the weighting factor calculated by the PCA resulted in a normalized catchments-clustering. This catchments-clustering is added to a geographic information system and results in a susceptibility map (Fig. 8).

Size Factor and Relief Factor also show a correlation with the volumes of debris flow deposits (Fig. 9ab). Sediment volumes of less than 6,000 m$^3$ were supplied from small, low-hierarchical catchments (low values of Size Factor, minor than 0.25), whereas from large, high hierarchical catchments (high values of Size Factor, greater than 0.75) volumes were higher than 6,000 m$^3$. On the other hand, sediment supply was lower in catchments with high mean slopes values and high topographic

gradients (high-Relief Factor, greater than 1.75) instead of catchments with values low slopes and low topographic gradients (low-Relief Factor, minor than 1.25). Therefore, topographic attributes involved in both conditioning-factors are also significant for debris flow deposit volumes.

## 5.3 Catchment lithological attributes

One hundred and sixty Schmidt Hammer (SH) rebound values were measured in Geo1, 240 in Geo2, 280 in Geo3 and 160 in Geo4. Data outside the standard deviation at each station were excluded to eliminate data that could be affected by operator handling errors. Whisker and box plots show the distribution values for each geological group (Fig. 10). The values of the 50-percentile (Q2) and the interquartile range (IQR=Q3 - Q1) represent the dispersion of values between 75 and 25 percentiles.

We used the normalized values of the Mean Schmidt Hammer (MeanSHn) and the Interquartile Range of Schmidt Hammer

(IQRSHn) to evaluate the strength of the unaltered rocks and the degree of weathering of the rocks within the catchments. The data used is available in the supplementary data 2. MeanSHn and IQRSHn resulted from the addition of SH mean and SH IQR values for each geological group weighted by their percentage of area for each catchment (GeoX).

$$MeanSHn = (Geo1 * 43) + (Geo2 * 49) + (Geo3 * 41) + (Geo4 * 25) + (Geo5 * 10) \tag{4}$$

$$IQRSHn = (Geo1 * 8) + (Geo2 * 6) + (Geo3 * 10) + (Geo4 * 10) + (Geo5 * 1) \tag{5}$$

MeanSHn and IQRSHn show a weak correlation in both catchment' types (49 catchments that generated debris flows and 75 that did not) (Fig. 6b). Furthermore, no correlation emerged between MeanSHn-IQRSHn and Size-Relief Factors (Fig. 6c-f). The percentage of catchments that generate debris flows does not vary significantly in relation to the range of MeanSHn and IQRSHn (Fig. 7cd). The latter suggests that unaltered-rock strength represented by MeanSHn and the degree of weathering

or the degree of fracture of the rocks represented by IQRSHn of geological units is not significant in the generation of debris flows for the catchments impacted by the March 2015 storm event. Moreover, MeanSHn and IQRSHn do not show a clear correlation with the volumes of debris flow deposits (Fig. 9cd).

## 6  Discussion

### 6.1  Conditioning-factors for the generation of debris flows

The results show a selective generation of debris flows from the tributary catchments within the Huasco Valley during the March 2015 storm under similar amounts of rain. According to the results, the selective activation of the tributary catchments could have been related to the topographic attributes that controlled the entrainment of sediments from the drainage network towards the alluvial fans during the March 2015 storm. On the contrary, the lithological attributes of the tributary catchments

do not show any correlation with selective activations. The selective activation of catchments has also been reported for El Elqui river valley situated 130 km south of the study area (Vergara et al., 2018). Vergara et al. (2018) also highlights that the intensity of the 1-h peak storm precipitation had a low significance in the distribution of debris flow generation during the storms. The results of the logistic regression model in El Elqui river valley (Vergara et al., 2018) support that the prediction of
high discharge events does not depend on the accumulated rainfall. These selective activations have been previously reported in other Andean catchments (Colombo, 2010; Lauro et al., 2017), which would be explained by different coupling status between tributary catchments and trunk valleys (Fryirs et al., 2007; Mather and Stokes, 2017).

Previous studies in geomorphology had shown that the measurement of rock strength can be used to indirectly determine weathering rates and to predict volumes and grain-size supply within the catchments (Roda-Boluda et al. (2018) and references
therein). Both approaches assist in evaluating the rate of debris production to the hillslopes and debris distributions in the catchments (Bovis and Jakob, 1999). However, the measurement of rock strength in El Huasco river valley does not show any correlation with the selective generation of debris flows in the tributary catchments during the March 2015 storm. We propose, based on the field observations and in the statistical results that the low significance of lithological attributes in the debris flow activation is due to the fact that debris supply from hillslopes is not a significant source of sediments during storms. Field
observations in this study, and also reported by Cabré et al. (2020), show that that the main source of debris flows during the March 2015 storm is sediments stored in the drainage-networks. The large volume of sediments in drainage-networks is enhanced by the low-frequency of extreme storm events that usually impact the area (1 event/ 200-40 years (Ortega et al. (2019) and references therein) that facilitates the storage of sediments in inter-storm periods and thus prepare these channels to yield sediment downsystem at any extreme storm event that impacts the area. This geomorphological 'behavior' is described
in the literature as transport-limited catchments (Bovis and Jakob, 1999) which is characteristic of arid landscapes and this is independent of the local lithological variability.

In El Huasco River valley, landslides during the march 2015 storm were rare on the debris-mantled hillslopes (Cabré et al., 2020). Recent studies to assess debris flow generation show that shallow debris-mantled failures on the hillslopes are not required to trigger debris flows in arid catchments during intense and low frequency storm events (Vergara et al., 2018).
Hillslope stability is favored by the transport-limited conditions characteristic of arid catchments. In addition, the transport limited condition favors the storage of sediment in the alluvial channels, where sediment entrainment towards tributary junction alluvial fans due to runoff can occur at any storm that affects the area (Coe et al., 2008; Kean et al., 2013). The latter is in agreement with our field evidence that considers the available debris in the alluvial channels as the main entrainment source areas of sediment that were able to generate debris flows during the March 2015 storm event in El Huasco River valley. In that
direction, similar interpretations were deduced based on field observations in catchments situated 300 km north of the study area, where flashfloods were reported as the main erosion agents in the alluvial channels while hillslopes remained surprisingly stable during the March 2015 storm event (Wilcox et al., 2016).

Extensive gully formation caused by runoff on the hillslopes in the study area have been reported during the March 2015 storm. These observations are consistent with previous reports that have quantified that gullies contribution is 50 to 80% of
the overall sediment yield induced by rainfall in arid zone (Poesen et al., 2003) and the landslides on hillslopes have little

contribution. In fact, landslides on hillslopes are related to the largest rainfall events worldwide (e.g., Coppus and Imeson, 2002; Berti et al., 2012) in monsoon impacted zones (e.g. Marc et al., 2018) and rainfalls of southern Chile that exceed the accumulated rainfall during the March 2015 event in the Atacama Desert by ten times (Jordan et al., 2019). Debris flows, hyper-concentrated flows and mud flows during intense rains in the central and southern Chile are generally linked to an increase

in the pore pressure of the surficial loose debris layer that results in shallow landslides on the hillslopes(e.g., Sepúlveda and Padilla, 2008). However, in view of the result presented in this work, stream-networks would be the main source of sediment during a storm in the arid Andes of northern Chile. These stream-networks, including gullies, are previously filled prior to the storm with sediment entrainment from the gravitational landslides and rockslides of hillslopes during the inter-storm period. This mechanism is efficient only when the topographic conditions of the drainage-networks allow it.

We have shown that topographic attributes play an important role in the generation of debris flow and distribution of debris flows during extreme storm events. Consequently, the largest tributary catchments, with highly developed drainage-networks and relatively low mean slopes, were activated during the March 2015 event. Large volumes of sediment stored in drainage-networks resulted in various debris flows during the storm that yielded large volumes of sediment to the outlets (Fig. 11). In contrast, small, steep catchments, with low developed drainage-networks, store fewer volumes of sediments in the channels

and, therefore, during extreme storms, generate flows with lower sediment to water concentrations. The high altitude of the zero-isotherm during the March 2015 storm explains the abundant generation of debris flow in the studied area. A larger area with effective water capture results in extensive runoff impact in large tributary catchments where it generally snows at the head of catchments.

### 6.2   The role of individual extreme storm events in erosion rates in the southernmost Atacama Desert

In this work, a mean erosion of 1.3 mm is calculated for the entire study area in El Huasco River valley during the March 2015 storm based on the volumes of the debris flow deposits in the alluvial fans. The volumes calculated from the debris flows deposits in the alluvial fans have an uncertainty of -34% and +46% and these volumes are the minimum values of erosion in the catchments due to the loss of sediment due to the river toe-cutting of alluvial fans. Therefore, volumes of approximately 1 million $m^3$ and mean erosion of 1.3 mm during the March 2015 storm in El Huasco river valley should be considered with

caution, and rather their order of magnitude should be considered. Limitations to study erosion associated with an individual storm event include the lack of pre-event high resolution topography (e.g. available high resolution digital elevation models; DEM's). The lack of detailed topographic surveys before the storm makes it difficult to quantify the volume balance of the difference DEM's (DOD's) to represent the geomorphic change caused by gullies and rills on the hillslopes and erosion in the alluvial channels (e.g., Cavalli et al., 2017). The sediment eroded and transported downsystem through the main rivers might be

recognized in the concentrations of suspended sediment measured at river gauges. However, these records also underestimate the maximum discharge associated with the storm because the gauging stations are commonly damaged by flashfloods.

The contribution of an individual extreme storm event to long-term erosion rates should only be considered for data spanning large areas because focused assessments may not be representative. In this work, the study area impacted by this storm of March 2015 is 1,500 $km^2$. A return of 116 years can be proposed for storms like the March 2015 as an average for the southern

Atacama Desert during the last 5,500 years based on Ortega et al. (2019). The proposed return time for the southern Atacama Desert is similar to the 100 years return time estimated by Houston (2006) on El Salado River, a tributary of El Loa Valley in the Altiplano. An erosion rate of $10^{-2}$ mm/yr could be estimate based on the mean erosion of 1.3 mm measurement during the March 2015 storm by this work and the return time of this type of events from the paleoclimatic record of 1 event each 100 years. This value agrees in order of magnitude with the Holocene erosion rates in the Huasco River Valley calculated from the concentration of Terrestrial Cosmogenic Nuclides in stream sediments and Plio-Quaternary erosion rates calculated from missing volumes of the deep-valleys incised below Miocene pediments and paleo-valleys (0.03-0.08 mm/yr calculated by Aguilar et al. (2011) and Aguilar et al. (2014) respectively). These two independent proxies of long-term denudation show a great significance of erosion linked to extreme storms on a scale of $10^6$-$10^4$ years. The latter are in agreement with a significant geomorphological contribution to the millennial erosion rates of extreme storm events deduced from the comparison of erosion rates at different timescales in arid and semiarid mountainous ranges (Kirchner et al., 2001; Kober et al., 2013; Carretier et al., 2013).

The occasional humid spells reported for the Mid-Holocene (8-4 ka) (Grosjean et al, 1997) or stormy conditions during this period (Tiner et al., 2018) produced the runoff necessary to entrain sediment from the alluvial channels and hillslopes within tributary catchments of the Atacama Desert. This has been evidenced in the Holocene stratigraphy of the alluvial fans of El Huasco river valley by a number of cohesive debris flow layers and radiocarbon age, interpreted as a result of episodic high-water discharge events during the last 8 ka BP (Cabré et al., 2017). Therefore, storm conditions and high sediment discharge occurred at least after 8 ka BP in the Atacama Desert. The increase of layers with coarse sediment in lacustrine and marine archives during the last 5,500 years BP is interpreted as high sediment discharge events associated with more frequent extreme storms in the southern Atacama Desert (Rein et al., 2005; Tiner et al., 2018; Ortega et al., 2019). Recurrence time decreased from 1 event/210 yr towards 1 event/ 40 yr in the last 1,000 years BP in the southern Atacama Desert (Ortega et al., 2019). However, the increase of coarse-sediment layers could be showing more summer-austral storms, with a high zero-isotherm altitude as the March 2015 storm. Therefore, this sedimentary record does not necessarily show changes of inter-decennial storm-return time linked to ENSO as recently interpreted by Ortega et al. (2019).

# 7 Conclusions

The generation of debris flows in the arid valleys of the southernmost Atacama Desert is controlled by the amount of available sediments stored within the catchments. The efficiency of the catchments for storing sediments depends in their topographic attributes that ultimately determine if the catchments are activated or not during extreme storm events. The sediment is stored during the inter-storm periods and, therefore, is susceptible to being transferred by debris flows during any extreme storm that impacts the area. Drainage-networks are the main entrainment sediment zones during extreme storms, while the supply of sediment from landslides on hillslopes is not observed for the generation of debris flows in these catchments during extreme storm events. The instantaneous increase in runoff during a storm and the entrainment of sediment has been associated to the high altitude of the zero-isotherm and the presence of storm cells. Evaluations of the susceptibility to debris flow in arid

catchments of the Atacama Desert that incorporate the topographic classification of catchment-clustering as predictors could be easily implemented avoiding extensive field-work observations.

Recently, there has been some consensus that storms like the March 2015 in the Atacama Desert occur on average every 100 years (Ortega et al., 2019). Therefore, the erosion rates associated with these storms are of the order of $10^{-2}$ mm/year in El Huasco river valley if we consider the erosion measurements after the March 2015 storm. The order of magnitude of erosion is the same as the calculated long term erosion rates in the Huasco River valley (Aguilar et al., 2014). This indicates that these storms have a significant influence on the erosion and evolution of these arid fluvial systems of the Atacama Desert. However, the influence of different surface processes on sediment preparation in the period between storms has not yet been characterized. New direct data, including quantitative geomorphological analysis of erosion, sedimentation and soil formation, are required to quantify and improve the knowledge of the rates, routes and magnitudes of each of the processes involved in the landscape evolution of the Atacama Desert.

Susceptibility assessment to the generation of debris flow and the estimation of the return time of large-scale alluvial episodes should be included in studies of hydro-meteorological hazard in populated areas (Wilford et al., 2004). The susceptibility of debris flow generation in the southern Atacama Desert is linked to the storage capacity in a drainage network during the inter-storm period. Furthermore, susceptibility is linked to the percentage of the catchment below the zero-isotherm altitude during the storm. The unusual altitude of the zero-isotherm above the topography during the March 2015 storm explains the lack of similar sedimentary behavior in the historical records and the extensive runoff of debris flows from the sediment source. However, alluvial episodes such as March 2015 control the evolution of the landscape of the fluvial systems of the Atacama Desert. The return times of 1 event every 100 years are adjusted to the millennial erosion rates. Considering the altitude of the zero-isotherm during storms in debris flow susceptibility assessments is essential because recent literature has predicted its progressive increase in mountainous areas of Chile (Carrasco et al., 2005; Boiser et al., 2016) and worldwide (Mountain Research Initiative EDW Working Group, 2015).

*Data availability.* 1. Measurement of debris flow deposits and volume calculation

*Data availability.* 2. Morphometric and geological features of catchments

*Competing interests.* The authors declare no competing interests.

*Acknowledgements.* This work is supported by the Basal Project of the Advanced Mining Technology Center financed by CONICYT Project AFB180004, Government of Chile. This work is based in the undergraduate thesis of V. Fredes and is part of the PhD thesis of A. Cabré

supported by the Chilean Government funded by CONICYT + PAI/ Concurso nacional de tesis de doctorado en el sector productivo, 2017 Folio (T7817110003). We gratefully acknowledge R. Riquelme, A. Mather, M. Lara and S. Sepúlveda contributions in these theses. We also acknowledge Planet Team grant (www.planet.com) given to A. Cabré. This manuscript result of valuable contributions during early publication in Natural Hazards and Earth System Sciences Discussions (NHESSD). Special thanks to the referees M. Mergili, T. Jordan and L. García, and another colleague for your participation in the early version of this manuscript.

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

**Table 1.** Table that summarizes the main characteristics of sedimentary facies of debris flows deposited during the storm in the tributary-junction alluvial fans (Cabré et al., 2020)

| Facies type | Description | Interpretation |
|---|---|---|
| Thick lobes matrix-supported gravels. | Lobes of sediment with variable widths (1 to 100 m) on the pre-existent fan surfaces. Thickness ranges between 30 and 100 cm. Pressure ridges on the lobe fronts include boulder accumulations interlocked, levees in the both sides of the channel can be observed along fan apex area. | Cohesive debris flows with high density and low viscosity originated from pulses of sediment that correspond to the first stages of sedimentation. The loss of confinement of flows, the loss of the water content, and the change of slope led to the sedimentation of this facies on the uppermost portion of the fans. |
| Thin lobes matrix-supported gravels. | Multiple stacked narrow lobes (<25 m) with thickness ranging from 10 to 30 cm. The lobes are narrow with length:width ratio typically between 1 and 4 with cobbles up to 15 cm. The matrix includes silty muds up to 5–12% and granule accumulations that present clast-supported fabric. | Diluted debris flows that cover the previus facies with negligible erosion. The transient from cohesive to more diluted facies are related to differences in water to sediment ratios. The morphology of these multi-lobe facies is related to multiple surges during the rainstorm event. |
| Arcuate push ridges and longitudinal lobate bars. | This facies only occur into and near the active channel. Present moderate thicknesses ranging from 5 to 30 cm depending on whether they are linear or lobate bar-forms. These facies have low matrix content and consist mainly of clasts up to 300 cm in the channel beds and up to 20 cm in the arcuate push ridges. | Cohesion-less debris flows. The arcuate shape of the flow deposits with openwork boulders represents push ridges as a consequence of new surges. The presence of boulders in the channels may be either related to the transport processes of previus facies, where, the matrix might have been washed out by greater water:sediment ratios. |
| Depositional bars. | Moderate depositional relief (<50 cm) that consist entirely of pebbles, cobbles and boulders forming elongated shape either in the feeder channels or in the lobes at the toe showing a radial pattern. | Hyper-concentrated flows. These facies are interpreted as fluvial longitudinal bars associated with the removal of previus facies. The low amounts of fine sediments suggest a continuous washing provided from the catchment. Many erosive features suggest the turbulent behaviour of the flows and provide space to fine-grained and sheet-like sands. |
| Fine-grained and sheet-like sands. | Sand bodies and granules, which fill channels or scatter on the active depositional lobes. Sheets vary between 1 and 2 cm in thickness and are well stratified. Matrix composed of mainly medium to fine sands, with minor silt and mud (12%). | Presence of sheet-like sands indicates tractional flow, and the lower amounts of fine sediments are attributed to the flushing of clean waters. They reflect a pulse of more water-rich flows through the system, which is common after the initially available sediment has been flushed out by the earlier part of the storm. |
| Fine silty/muddy sheets. | Fine sheets of silty/muds with mud-cracks, wood fragments and vegetation. | Facies are deposited above where the active depositional lobe of the tributary-junction alluvial fans dams the main river, creating ephemeral palustrine conditions. These facies contain mud-cracks that are developed on the surface by evaporation. |

**Table 2.** Table that summarizes the percentages of catchments prone to generate debris flows and presents thresholds of topographic attributes that contribute within each factor.

| Size Factor | Catchments (%) | Area (km$^2$) | Length (km) | Strahler Order |
|---|---|---|---|---|
| 0.05-0.25 | 18 (4 of 22) | <1 | <2 | 1-2 |
| 0.25-0.75 | 39 (30 of 74) | 1-7 | 2-5 | 2-4 |
| 0.75-1.50 | 57 (13 of 23) | >7 | >5 | 4-6 |
| Relief Factor | Catchments (%) | Slope ($^o$) | Melton (km/km$^2$) | Gradient (km/km) |
| 1.75-2.00 | 9 (1 of 11) | >30 | >1.1 | >0.6 |
| 1.75-1.25 | 33 (15 of 46) | 1-7 | 1.1-0.7 | 0-6-0.4 |
| 1.25-0.25 | 50 (31 of 62) | <26 | <0.7 | <0.4 |

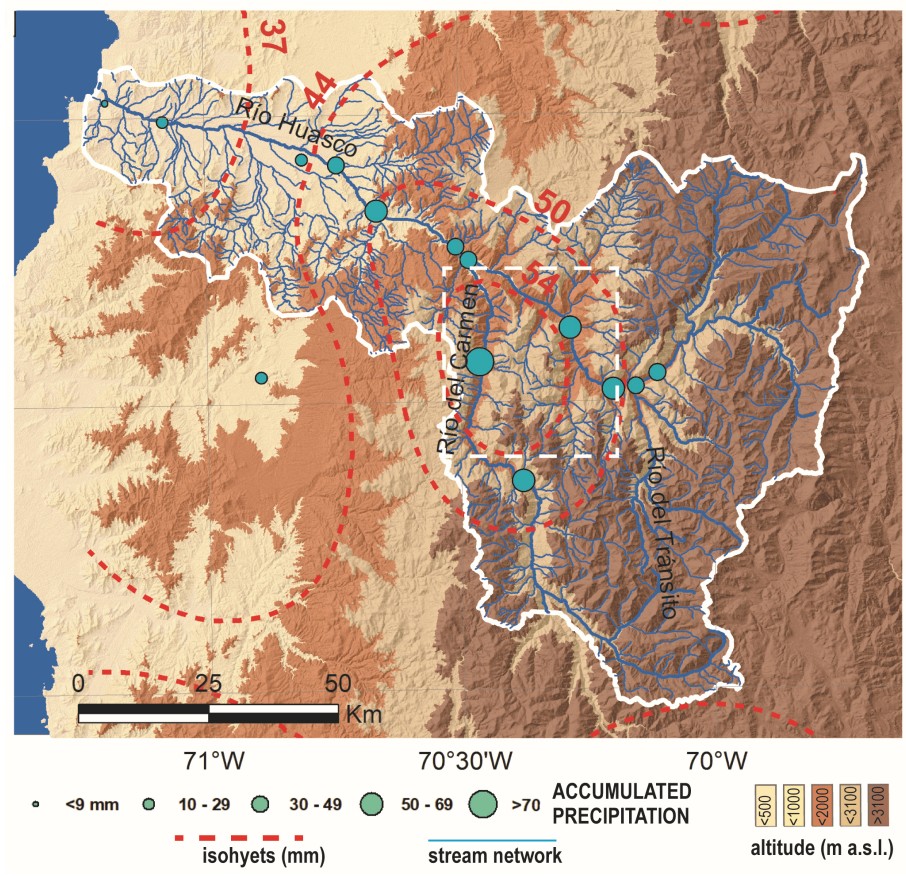

**Figure 1.** Topography extracted from a Digital Elevation Model Courtesy NASA/JPL-Caltech with a nominal spatial pixel resolution of 30 x 30 m (https://asterweb.jpl.nasa.gov/gdem.asp) and modeled hydrology of the Huasco River Valley. Dotted red lines show the isohyets of daily accumulated rains during the March 23–26th storm by kriging interpolation of meteorological data from Dirección General de Aguas - Chile. Cyan points show the accumulated rainfall in each meteorological station during the March 2015 storm. Dashed-line white box show the studied region.

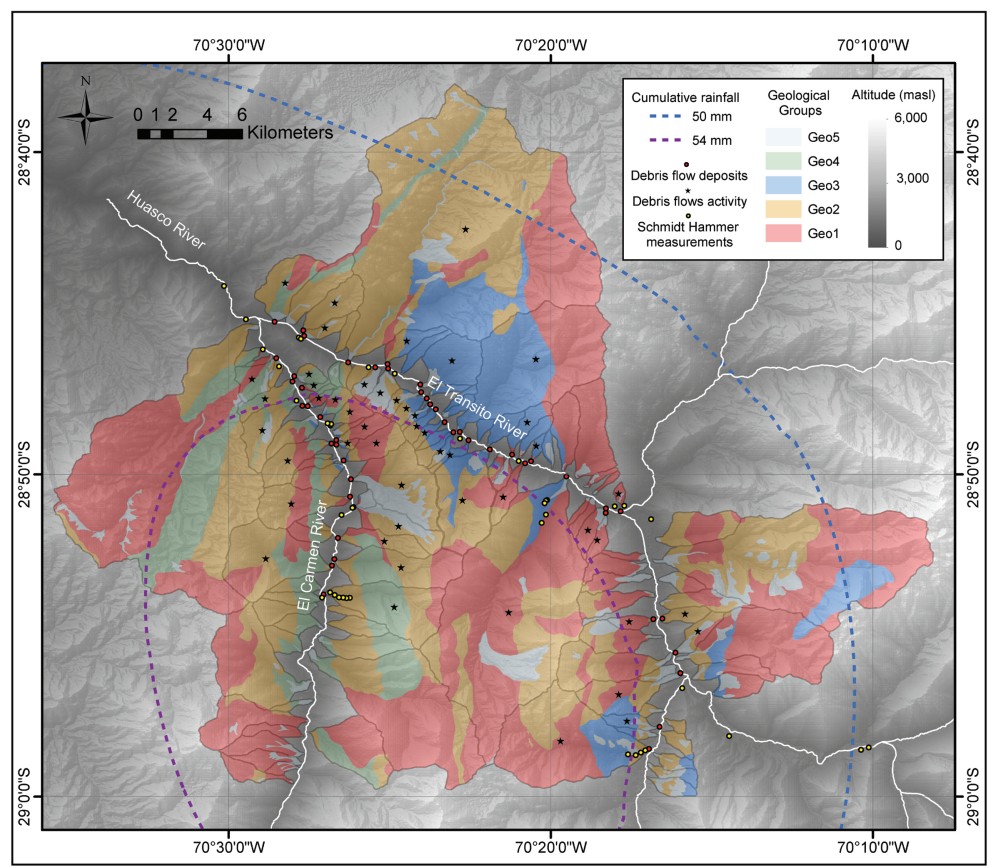

**Figure 2.** Map that shows the distribution of Geological Groups defined in El Huasco river valley after Salazar et al. (2013) with the location of Schmidt Hammer stations. The map includes the distribution of debris flow deposits during March 2015 and isohyets of accumulated rains created by kriging interpolation of meteorological data during the storm. Topography extracted from a Digital Elevation Model Courtesy NASA/JPL-Caltech with a nominal spatial pixel resolution of 30 x 30 m (https://asterweb.jpl.nasa.gov/gdem.asp).

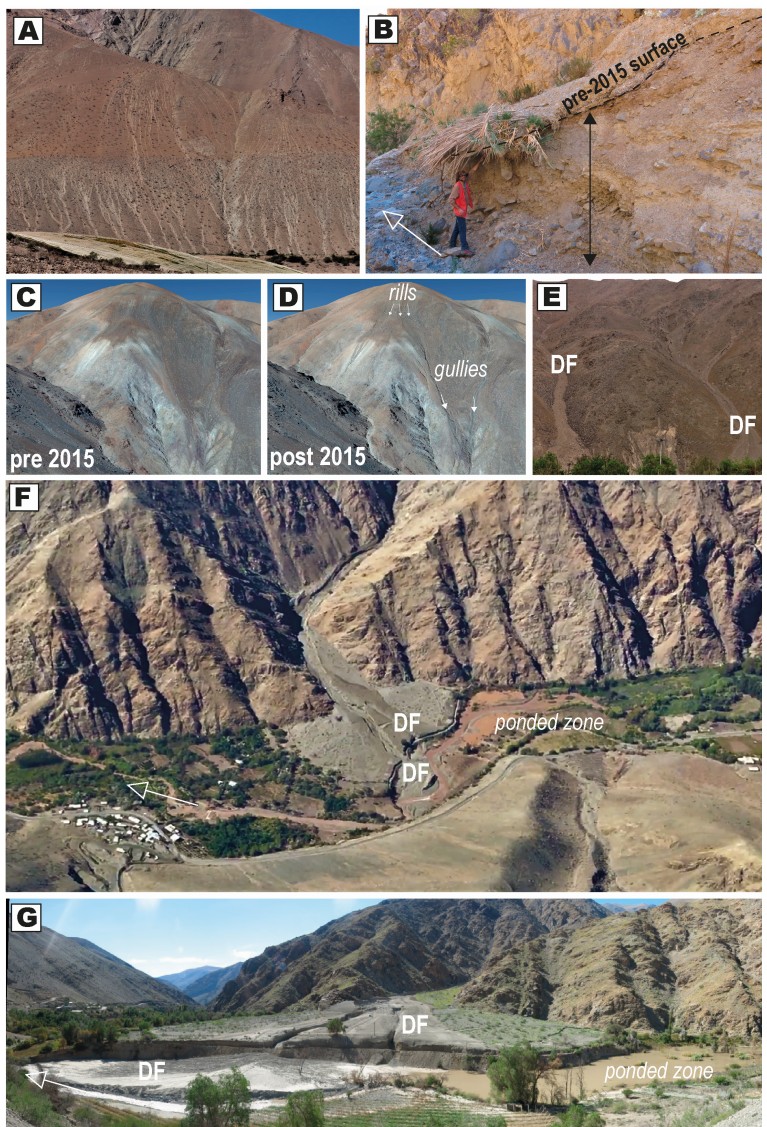

**Figure 3.** (A) Field evidence of high-density rill formation during the March 2015 storm on hillslopes of the Huasco River Valley. (B) Example of a deeply incised alluvial channel within a tributary catchment. Dashed line indicates pre-March 2015 storm surface with covered vegetation lying down overlain by a 30 cm layer of debris flows (DF) deposit triggered by the storm inand subsequently incised by more diluted flows. Person for scale is 180 cm tall. White arrow indicates the flow direction downwards. (C) and (D) indicate pre and post landscape configuration showing rills, gullies and debris flows. (E) Debris flows (DF) with lateral levees deposited within the tributary catchment alluvial channels with subsequent incision. (F) Oblique aerial view of debris flows deposits on a tributary junction alluvial fan and the upstream zone which was ponded by the damming with the distal depositional lobe. G. Front view of the tributary-junction alluvial fan of F).

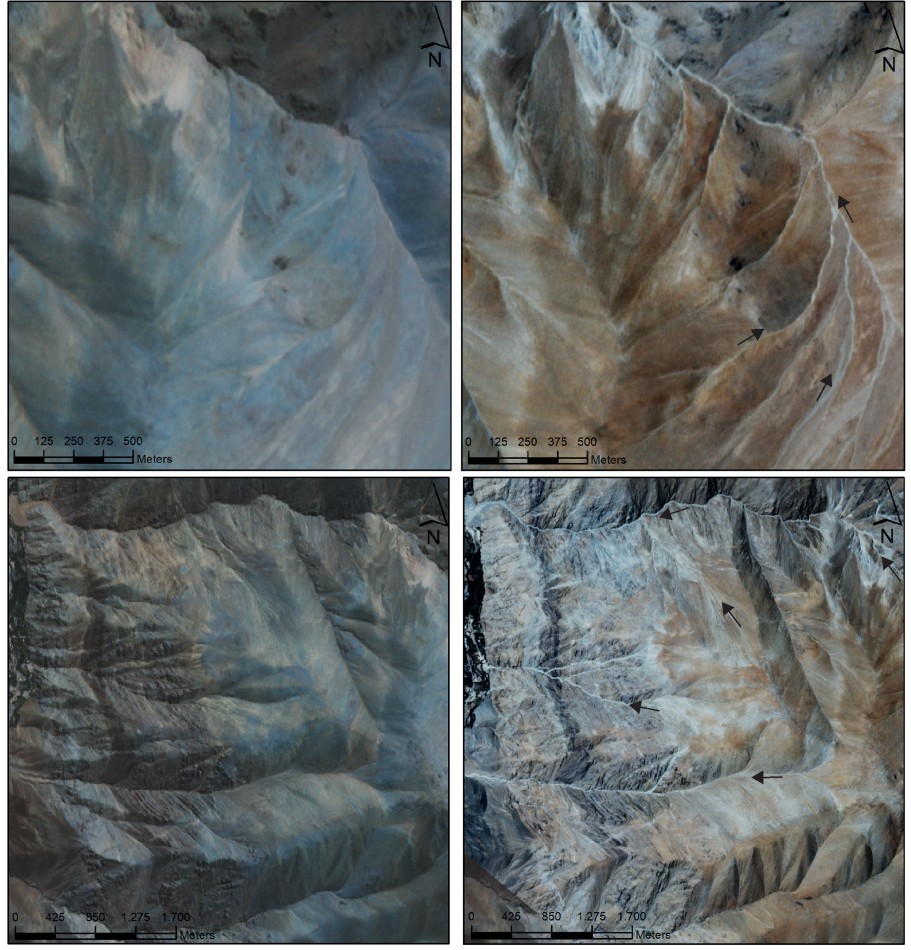

**Figure 4.** Optical imagery retrieved from Planet Team (2017) before and after March 2015 storm. Left images are before the March 2015 storm and right images are after. Arrows indicates different physical evidence of erosion processes due to the event and gully presence after March 2015 storm. The general color tone change merely indicates different sun illumination.

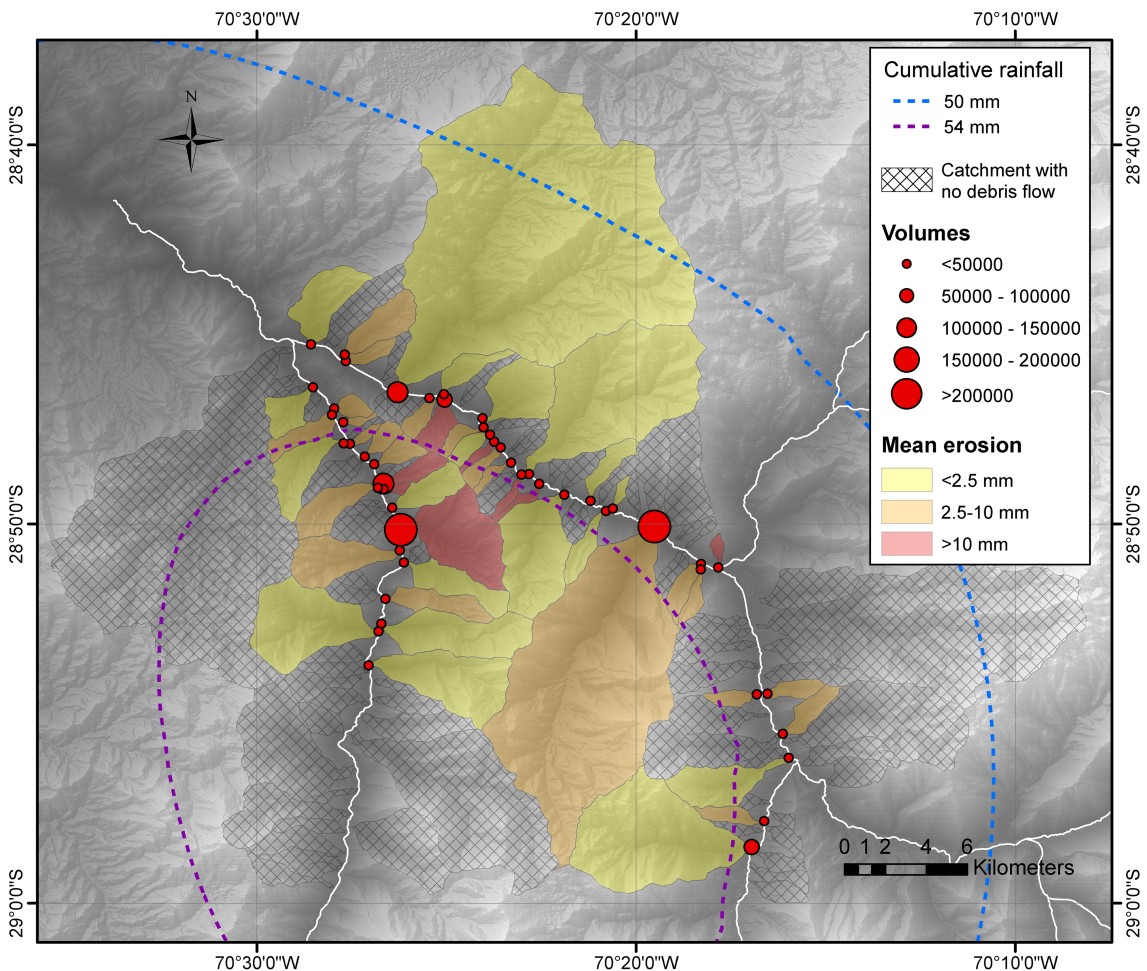

**Figure 5.** Distribution and volumes of debris flow deposits during the March, 2015 storm event in El Huasco river valley and identification of catchments where these flows were generated. Map shows the distribution of calculated mean erosion of tributary catchments and isohyets of accumulated rains by kriging interpolation of meteorological data during the storm. Topography extracted from a Digital Elevation Model Courtesy NASA/JPL-Caltech with a nominal spatial pixel resolution of 30 x 30 m (https://asterweb.jpl.nasa.gov/gdem.asp). The area of the map is the Dashed-line white box of Fig. 1

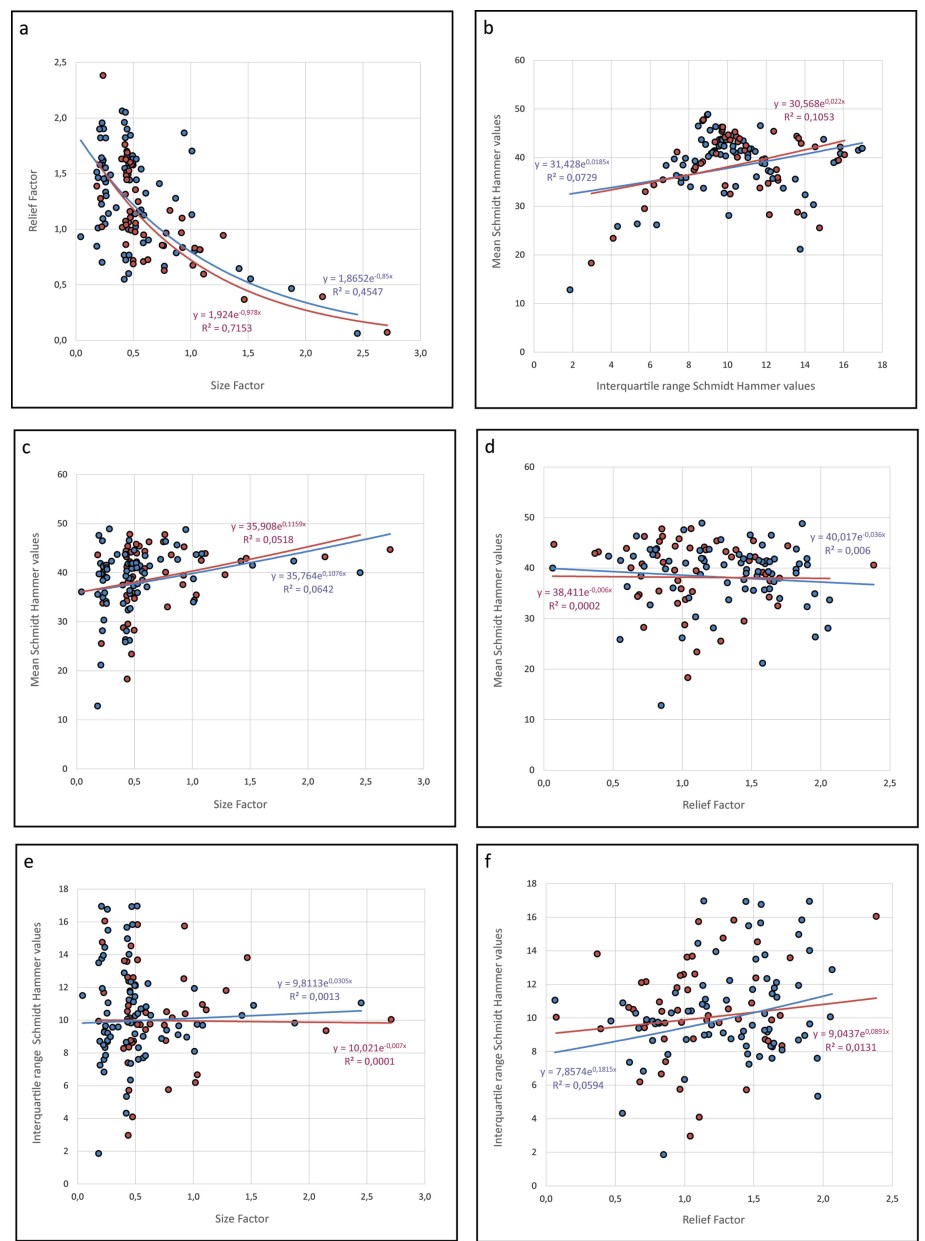

**Figure 6.** Graphics show relationship between different studied factors in catchments with debris flow generation in red and non-generation in blue. (a) Relationship between Size Factor and Relief Factor. (b) Relationship between Mean and Interquartile range of Schmidt Hammer values. (c) Relationship between Mean Schmidt Hammer values and Size Factor. (d) Relationship between Mean Schmidt Hammer values and Relief Factor. (e) Relationship between Interquartile range of Schmidt Hammer values and Size Factor. (f) Relationship between Interquartile range of Schmidt Hammer values and Relief Factor. In all graphics dotted box indicate ranges with enough data to statistically analysis (greater than 5 catchments for range).

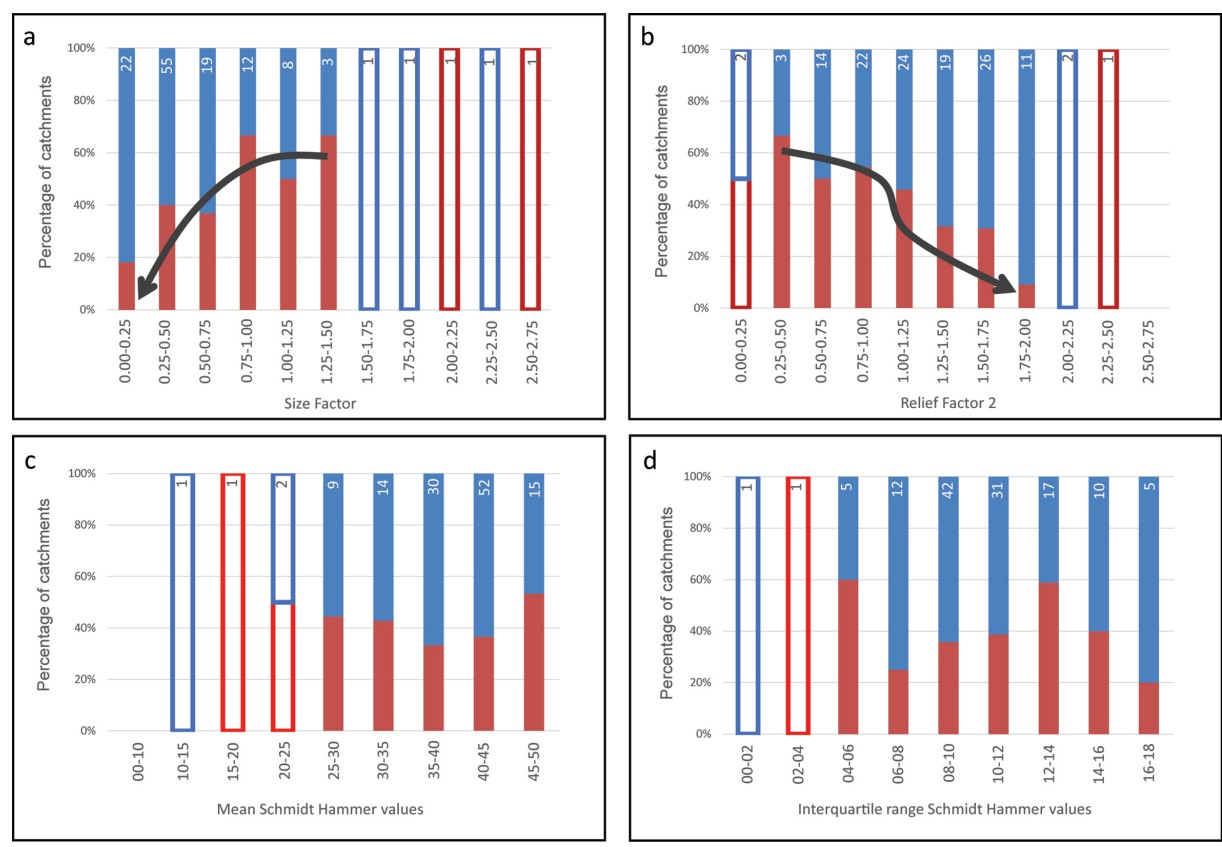

**Figure 7.** Histograms with the relative percentage of catchments with debris flow generation in red and non-generation in blue considering range of values for: (a) Size Factor, (b) Relief Factor, (c) normalized mean Schmidt Hammer values and (d) normalized interquartile range Schmidt Hammer values. Each bar includes a number indicating the quantity of analyzed catchments. Open bars indicate ranges without enough data to justify a statistical analysis (fewer than 5 catchments for range). Arrows indicate a trend in the histograms

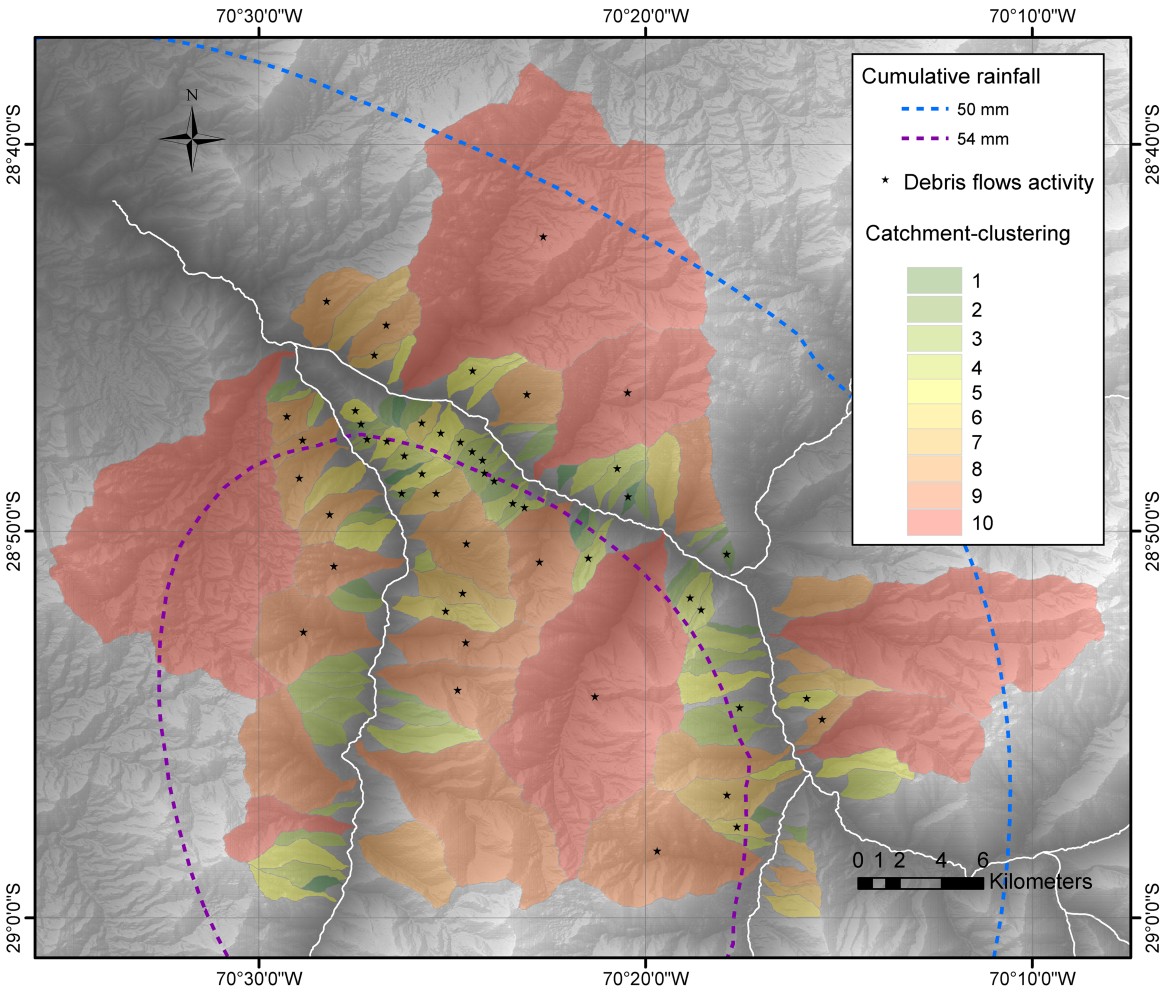

**Figure 8.** Map of the normalized classification of catchment-clustering considering the six topographic attributes (Area, Length, Strahler Order, Slope, Melton ratio and Relief ratio) involved in conditioning-factor of debris flow generation. Topography extracted from a Digital Elevation Model Courtesy NASA/JPL-Caltech with a nominal spatial pixel resolution of 30 x 30 m (https://asterweb.jpl.nasa.gov/gdem.asp).

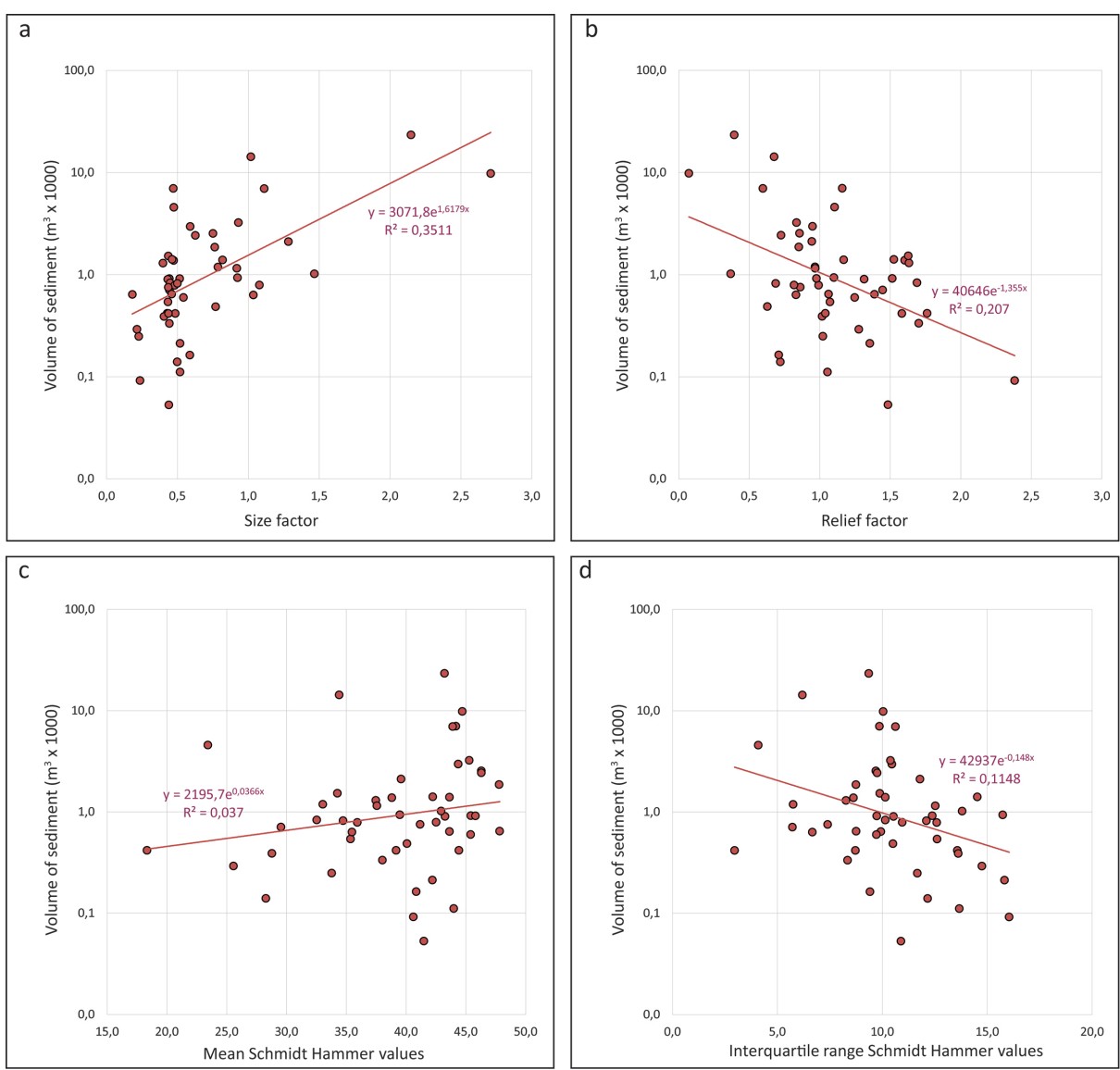

**Figure 9.** Graphics showing the relationship between different studied factors in catchments with sediment volumes of debris flow deposits. (a) Relationship between Size Factor and sediment volumes. (b) Relationship between Relief Factor and sediment volumes. (c) Relationship between Mean Schmidt Hammer values and sediment volumes. (d) Relationship between interquartile range Schmidt Hammer values and sediment volumes.

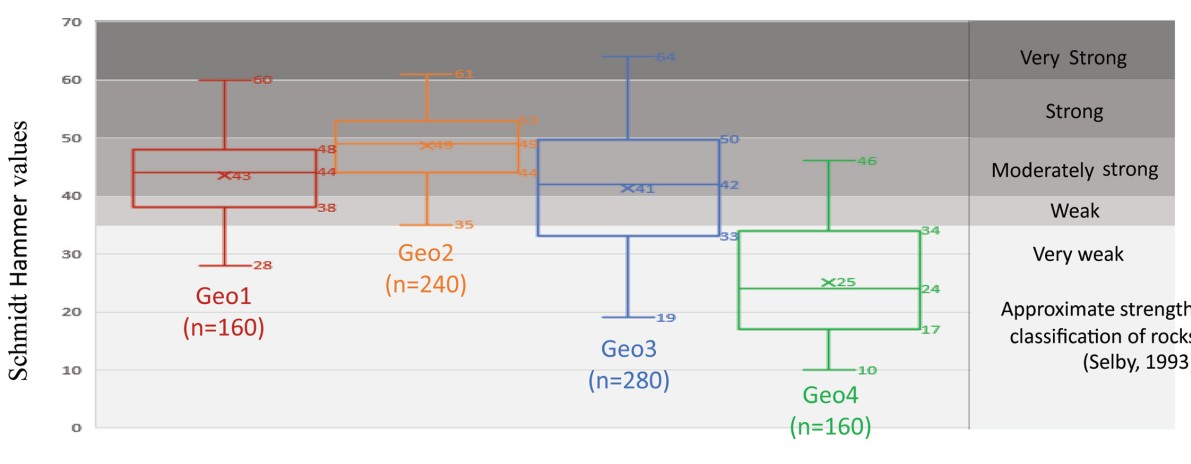

**Figure 10.** Whisker and box plots of Schmidt Hammer values for geological groups of the studied zone where n represent the number of measurements. Graphic show maximum, minimum and mean values for each geological group and also the values of 75th, 50th and 25th percentiles for each group. Graphic show the approximate strength classification of rocks by Schmidt Hammer values after Selby (1993).

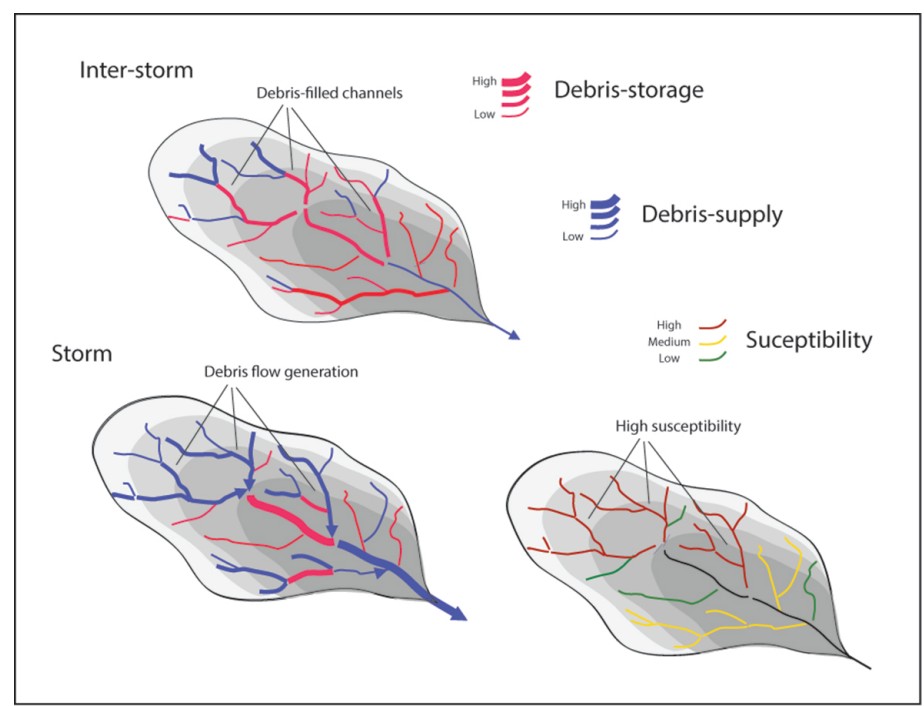

**Figure 11.** Conceptual model of the relation between hydraulic capacity during the intense long-lived rainfalls of an extraordinary storm and debris storage at previous time spam that explain different levels of debris flow susceptibility depending on catchment topography.