# Peer review of "Erosion after an extreme storm event in an arid fluvial system of the southern Atacama Desert: an assessment of magnitude, return time, and conditioning factors of erosion and debris flows generation"

_Natural Hazards and Earth System Sciences, 2019_

## Referee Comment (RC1) · Martin Mergili (Referee) · 20 Aug 2019

The authors present an interesting piece of research on erosion related to debris flows in the southern Atacama Desert, Chile, during an intense rainstorm, concluding that sediment storage in the channels is much more important than sediment supply by slope processes. The work is certainly scientifically interesting and within the scope of the journal. The manuscript is largely well-structured and the illustrations are appro-

priate and very well-designed. Gramnmar and style are of reasonable quality at the beginning of the paper, but degrade significantly afterwards. Maybe also for this reason, there are some aspects which are not clear for me and where I suggest rewriting. Therefore I recommend moderate-major revisions. Please find my detailed comments and suggestions below.

Most importantly, carefully check grammar and style (maybe you can ask a native speaker to go through the text). Below I highlight some, but not all language issues I have identified.

Title: "erosion caused by debris flows": is it really the debris flows which caused the erosion, or is it the reverse way, or both?

P2, L32: "On the other hand ...": Something is wrong with the structure of this sentence, please reformulate.

P3, L4f: How do you know about the different rheologies? Analysis of deposits, interviews with witnesses, ...?

P3, L7: lectors -> readers

P5, L2: time return -> return time

P5, L11f: If no debris flows were reported, does it really mean that no sediment was delivered to the trunk valley? Even though you use a rather broad definition of debris flow, there might still be some fluvial sediment transport.

P3, L24: Rills and gullies are not processes, but landforms - better replace "processes" by "indicators".

P5, L22f: What is the difference between slope and gradient?

P5, L23: lineal -> linear

P6, L12: "is added" should be removed.

[Figure]

P6, L22-29: This paragraph should be moved to the section on the study area, as it represents some general information, not the results of the study.

P7, L16: "Andean catchments": the Andes are thousands of kilometres long, please be more precise ("arid Andes of northern Chile", or whatever is suitable).

P7, L21 and some other places in the manuscript: alluviated -> alluvial

P8, L5f: "high latitude rainfalls": please mention to which region you refer exactly (Patagonia?)

P9, L4: "first phase of risk study inhabited areas": I don't understand this formulation.

P9, L10: Only in Chile, or is it relevant in mountain areas worldwide?

P9, L21: The recurrence time decreased (it is the frequency which increased).

P9, L33f: "The integration ...": I do not understand this sentence, please reformulate.

P10, L27f: But how does the sediment get into the channels? On long (geological) time scales, hillslope processes probably play a role?

Fig. 5 and Fig. 8: It could be interesting to see the $R^2$-values in each of the diagrams.

In case the authors would like to discuss the one or the other issue, they should feel free to contact me at martin.mergili@univie.ac.at.

With best regards

Martin Mergili
* * *

---

## Author Comment (AC1) · 7 Oct 2019

Dear Martin Mergili,

We would like to thank you for taking the time to review our manuscript. We highly appreciate his suggestions and comments, which are helpful in improving the manuscript. We are going to include all the suggested modifications, particularly improve the quality of the English. We would like to answer two significant comments to clarify and maybe

start a discussion:

P3, L4f: How do you know about the different rheologies? Analysis of deposits, interviews with witnesses, ...?

[Reply] Rheologies was inferred by the analysis of deposits after March 2015 storm including sedimentology and geomorphological patterns in selected alluvial fans. This characterization are in the PhD Thesis of A. Cabré and in the manuscript entitled "Tributary-junction alluvial fan response to an ENSO rainfall event at El Huasco river watershed, northern Chile." which we submitted to Progress in Physical Geography (Cabré et al., submitted PPG).

P5, L11f: If no debris flows were reported, does it really mean that no sediment was delivered to the trunk valley? Even though you use a rather broad definition of debris flow, there might still be some fluvial sediment transport.

[Reply] Yes, effectively. Sure that sediment was delivered from all tributary affected by the storm, but only debris flows where deposited in the tributary-junction of 49 catchments. For these reason, we consider the volumes measured like a minimum of transference of sediment to the trunk valley (see line 15-21, page 10).

P10, L27f: But how does the sediment get into the channels? On long (geological) time scales, hillslope processes probably play a role?

[Reply] Very good question. Although it is not the subject of this specific work, we can advance that the filling of the canals develops rapidly in some years after the storm based on observations made in a subsequent storm occurred in May 2017. So, it is not necessary so much time, since the slopes have enough sediment to fill the canals in a short time, in particular from sediment stored in the "flat" slopes of the head of the catchments, mainly from upstream the knickpoints. But, we think that this is another manuscript that need more data (TCN and provenance studies).

Best regards

German Aguilar

---

## Referee Comment (RC2) · Teresa Jordan (Referee) · 11 Oct 2019

This is an informative paper that is appropriate to the focus of the journal "Natural Hazards and Earth System Sciences".

The manuscript contains numerous unclear passages, yet the major points are well developed and thoughtfully analyzed. I recommend that it be published after major revisions.

[Figure]

I have made numerous editorial suggestions within the pdf of the manuscript (attached). A few broader points are noted here.

1) The introduction lacks any statement of background information that would justify the authors' hypothesis that rock strength is a control on the rate of weathering or the generation of debris flows. Lacking that background, this reader was surprised to find that data are collected with a Schmidt Hammer, and that mean and IQR values of Schmidt Hammer data are considered to be potentially meaningful. Previous studies which make these assertions should be briefly described, including clarification of the rock types, climate zones, and topographic characteristics of the catchments from which the previous studies extracted their interpretations.

Related to the lack of background information, the reader does not know whether to treat the result that there is no correlation of SH mean and IQR to the phenomenon of debris flow generation as a surprising result (because it contradicts a body of published knowledge), or instead as a demonstration that the hypothesis was negated here and may likely also be incorrect in other locations.

2) The authors refer in the Discussion, section 4.1, to the generation of debris flows in tributary catchments as "random." I do not think that they have provided data which justify the statement that the phenomenon occurs randomly. In fact, it seems that their discovery that topographic features of the catchments are predictors of the spatial distribution of debris flows suggests that the phenomenon is not random.

3) In section 4.2, the concept of an "uncoupled" landscape is referred to. Nothing earlier in the paper provided an explanation of what the authors mean.

4) The caption to Figure 3 lacks vital information and guidance. At the least, it should be stated that left sides are "before" and right sides are "after". We also need to know whether the general color tone change is a physical evidence of erosion due to the March event, or if it merely indicates different sun illumination.

Please also note the supplement to this comment:
https://www.nat-hazards-earth-syst-sci-discuss.net/nhess-2019-239/nhess-2019-239-RC2-supplement.pdf

―――――――――――――――――――

[Figure]

**Supplement:**

[revised manuscript text omitted]

---

## Referee Comment (RC3) · Juan Luis García (Referee) · 25 Oct 2019

The paper by Aguilar et al. has as a main goal to asses the erosional impact during a single extreme rain event in southernmost Atacama Desert. The paper is well written and easy to follow. The structure is correct and conclusions are in line with the research goals and questions presented in the introduction. The paper provides new knowledge on the geomorphic dynamics in the southern Atacama, including quantitative data obtained both from the field and using remote sensing. The authors put their results in a wider context with direct impact in our understanding of the erosion rates in the Atacama and the main factors underlying them. To my knowledge, the approach of this work is unique and valuable for publication in NHESS.

Minor comments.

Line 17, p2: space between "by" and "terrestrial" Line 21, p3: you mean processes Line 3-4, p6: question: This statement refers to debris flows reaching the main valley, isn't? I mean, it probably there were debris flows within the catchment but no big enough to deliver sediment to the outlet alluvial fan? Line 14, p6: regarding positive correlation you mention: the higher the relief factor the steeper the slope within the catchment? therefore negative correlation with volumes of debris flows? Line 16, p8: erase ";" Line 24, p8: replace "did" by " does"

---

## Author Comment (AC2) · 31 Oct 2019

Dear Teresa,

We kindly appreciate your suggestions and comments on the manuscript entitled «Erosion after an extreme storm event in an arid fluvial system of the southern Atacama Desert: an assessment of magnitude, return time, and conditioning factors of erosion caused by debris flows». By including them the manuscript presents a greater clarity

and allows us to present a better and more useful contribution. I will now individually answer the main comments that you provided.

1) «The introduction lacks any statement of background information that would justify the authors' hypothesis that rock strength is a control on the rate of weathering or the generation of debris flows. Lacking that background, this reader was surprised to find that data are collected with a Schmidt Hammer, and that mean and IQR values of Schmidt Hammer data are considered to be potentially meaningful. Previous studies which make these assertions should be briefly described, including clarification of the rock types, climate zones, and topographic characteristics of the catchments from which the previous studies extracted their interpretations. Related to the lack of background information, the reader does not know whether to treat the result that there is no correlation of SH mean and IQR to the phenomenon of debris flow generation as a surprising result (because it contradicts a body of published knowledge), or instead as a demonstration that the hypothesis was negated here and may likely also be incorrect in other locations. »

[Reply] Measurement stations of Schmidt hammer were selected considering a statistic analysis of lithology, structural context, and geomorphology of the studied area. Details of this work are in the undergraduate thesis of Fredes (2016) (http://repositorio.uchile.cl/handle/2250/140357). In this analysis we take into account a background-review the geology 1:100.000 of Sernageomin (Salazar et al.2013) and geomorphological studies of two Ph.D. thesis of the Universite de Toulouse and Universidad Catolica del Norte (Aguilar, 2010; Cabré, 2019). We will show all the available data in the reviewed version of the manuscript. We will also cite more works that benefit from Schmidt hammer measurements to quantify resistance to rock weathering in catchments. Nevertheless, the validation of the Schmidt Hammer is far from being the focus of this paper.

2) «The authors refer in the Discussion, section 4.1, to the generation of debris flows in tributary catchments as "random." I do not think that they have provided data which

justify the statement that the phenomenon occurs randomly. In fact, it seems that their discovery that topographic features of the catchments are predictors of the spatial distribution of debris flows suggests that the phenomenon is not random. »

[Reply] We should have used "heterogeneous" rather than "random" to explain the different hydrological responses registered in relatively small catchments (<100km2) in this region of the Andes and therefore in neighboring tributary-junction alluvial fans. We will clarify this in the manuscript.

3) «In section 4.2, the concept of an "uncoupled" landscape is referred to. Nothing earlier in the paper provided an explanation of what the authors mean. »

[Reply] We refer to a landscape in a transient state with "uncoupled" surfaces, i.e. low degrees of connectivity within their catchments. We will clarify this passage.

4) «The caption to Figure 3 lacks vital information and guidance. At the least, it should be stated that left sides are "before" and right sides are "after". We also need to know whether the general color tone change is a physical evidence of erosion due to the March event, or if it merely indicates different sun illumination. »

[Reply] We will improve figure 3 to avoid confusion.

Best regards

German Aguilar

---

## Author Comment (AC3) · 31 Oct 2019

Dear Juan Luis, thank you very much for your comments and your appreciation for our manuscript. Here I include the answer to your questions. These will be considered in the corrected version:

« Line 3-4, p6: question: This statement refers to debris flows reaching the main valley, isn't? I mean, it probably there was debris flows within the catchment but no big enough

to deliver sediment to the outlet alluvial fan? »

[Reply] The percentage of catchments that generated debris flows were calculated considering if the flows reached the trunk rivers. It is very probable that was generated debris flows in other catchments and that did not reach the trunk rivers. We will clarify this in the corrected version.

« Line 14, p6: regarding positive correlation you mention: the higher the relief factor the steeper the slope within the catchment? therefore negative correlation with volumes of debris flows? »

[Reply] Indeed, there is an error that we will solve in the corrected version: The relationship is negative between the volume of sediments and the relief factor.

Best regards

German Aguilar

---

## Author Comment (AC4) · 18 Nov 2019

Dears reviewer and editor, following I include specific responses to your comments and corrections. Attached pdf file is the corrected manuscript include all modifications. Thank you very much for your great contribution in ours manuscript. Best Wishes German Aguilar

Specific Answers to Reviewer 1

Title: "erosion caused by debris flows": is it really the debris flows which caused the erosion, or is it the reverse way, or both?

[Reply] We change to « Erosion after an extreme storm event in an arid fluvial system of the southern Atacama Desert: an assessment of magnitude, return time, and conditioning factors of erosion and debris flows generation»

P2, L32: "On the other hand ...": Something is wrong with the structure of this sentence, please reformulate.

[Reply] Delete "On the other hand ..."

P3, L4f: How do you know about the different rheologies? Analysis of deposits, interviews with witnesses, ...?

[Reply] We change to « Analysis of deposits showed different rheologies of debris flows ranging from cohesive debris flows and hyper-concentrated flows to mud flows.»

P3, L7: lectors -> readers

[Reply] Change for "Readers""

P5, L2: time return -> return time

[Reply] Change for "return time

P5, L11f: If no debris flows were reported, does it really mean that no sediment was delivered to the trunk valley? Even though you use a rather broad definition of debris flow, there might still be some fluvial sediment transport.

[Reply] We include a threshold values of debris flow deposit: «Debris flows that reached the tributary junctions with the trunk valley and produced deposit greater to 500 m3 of sediment during the March 2015 event were reported in forty-nine of one hundred twenty-four catchments (Fig. 4).The remaining seventy-five catchments did not yield debris flows deposits greater to 500 m3 of sediments in the tributary junctions»

P3, L24: Rills and gullies are not processes, but landforms - better replace "processes" by "indicators".

[Reply] Change for "indicators"

P5, L22f: What is the difference between slope and gradient?

[Reply] Gradient is the inclination of length between tributary outlets and its more distant point of catchment. Slope refer to average slope within a catchment. We include modification to clarify:

P3, L28,29: « Area, Length (straight-line between tributary outlets and its more distant point), Maximum elevation, Gradient (inclination of length), Average Slope, ..... »

P5, L21: « Group 2 integrates the catchment relief and includes Average Slope (S), Gradient (G) and Melton ratio (M). »

P6, L10: « The six topographic attributes (Area, Length, Strahler Order, Average Slope, Gradient, and Melton ratio) »

P5, L23: lineal -> linear

[Reply] Change for "linear"

P6, L12: "is added" should be removed.

[Reply] Removed "is added"

P6, L22-29: This paragraph should be moved to the section on the study area, as it represents some general information, not the results of the study.

[Reply] We include a subsection "Study area" in the corrected manuscript that contain this paragraph. This section contain others information to answer comments of R2 (T. Jordan) about the lack of geological and geomorphological context.

P7, L16: "Andean catchments": the Andes are thousands of kilometres long, please be more precise ("arid Andes of northern Chile", or whatever is suitable).

[Reply] We change to « Debris flows, hyper-concentrated flows and mud flows in arid and semiarid Andes of northern Chile during intense rainfalls are usually linked to an increase in the pore pressure of the surficial loose debris layer generating a shallow-slide in the hillslopes of catchments.»

P7, L21 and some other places in the manuscript: alluviated -> alluvial

[Reply] Change for "alluvial" in the whole manuscript

P8, L5f: "high latitude rainfalls": please mention to which region you refer exactly (Patagonia?)

[Reply] We change to « .....and rainfalls of southern Chile that exceed the accumulated rainfall during the March 2015 event in the Atacama Desert by ten times....»

P9, L4: "first phase of risk study inhabited areas": I don't understand this formulation.

[Reply] We change to « Susceptibility assessment to debris flow generation must be evaluated in hydro-meteorological hazard studies in populated area. »

P9, L10: Only in Chile, or is it relevant in mountain areas worldwide?

[Reply] We include «..... and in mountain areas worldwide (Mountain Research Initiative EDW Working Group, 2015).»

Mountain Research Initiative EDW Working Group. Elevation-dependent warming in mountain regions of the world.Nature Climate Change volume 5, 424–430, https://doi.org/10.1038/nclimate2563, 2015.

P9, L21: The recurrence time decreased (it is the frequency which increased).

[Reply] Change for "decreased"

P9, L33f: "The integration ...": I do not understand this sentence, please reformulate.

[Figure]

[Reply] We change to « The time period of denudation rates calculated by Aguilar et al. (2014) is between 20 ka for sand and 12 ka for gravels. »

P10, L27f: But how does the sediment get into the channels? On long (geological) time scales, hillslope processes probably play a role?

[Reply] We think that long-therm geomorphological processes of catchments is not the subject of this specific work. We observed that the filling of some channel develops rapidly in some years after the storm based on observations made in a subsequent storm occurred in May 2017. So, it is not necessary so much time, since the slopes have enough sediment to fill the canals in a short time, in particular from sediment stored in the "flat" slopes of the head of the catchments, mainly from upstream the knickpoints. We think that this is another manuscript that need more long-therm data (TCN and provenance studies).

Fig. 5 and Fig. 8: It could be interesting to see the R2 -values in each of the diagrams. [Reply] We include the R2 of data in figures.

Please also note the supplement to this comment:
https://www.nat-hazards-earth-syst-sci-discuss.net/nhess-2019-239/nhess-2019-239-AC4-supplement.pdf
* * *

---

## Author Comment (AC5) · 18 Nov 2019

Dears reviewer and editor, following I include specific responses to your comments and corrections. Attached pdf file is the corrected manuscript include all modifications. Thank you very much for your great contribution in ours manuscript. Best Wishes German Aguilar

Specific Answers to Reviewer 2

1. Introduction

P2 Line 24-25: « On the other hand, meteorological stations and fluviometric stations suitable to measure hourly-rain intensity and sediment load yielding respectively, are scarce or lacking in the Atacama Desert.»

Comment R2: «The problem is compounded the lack or scarcity of meteorological ...»

Change: « The problem is compounded the lack or scarcity of meteorological and fluviometric stations suitable to measure hourly-rain intensity and sediment load yielding, respectively.»

P2 Line 28-29: « We have calculated the erosion within an area of 1,500 km2 based on volumes of debris flow deposits measured in alluvial fans after the storm.»

Comment R2: « if this 1500 km2 area is the white box in Figure 1, you should state that specifically.»

Change: « We have calculated the erosion within tributary catchments (whole area of 1,500 km2) based on volumes of debris flow deposits measured in tributary junction alluvial fans after the storm.»

Main comment: ÂńThe introduction lacks any statement of background information that would justify the authors' hypothesis that rock strength is a control on the rate of weathering or the generation of debris flows. Lacking that background, this reader was surprised to find that data are collected with a Schmidt Hammer, and that mean and IQR values of Schmidt Hammer data are considered to be potentially meaningful. Previous studies which make these assertions should be briefly described, including clarification of the rock types, climate zones, and topographic characteristics of the catchments from which the previous studies extracted their interpretations. Related to the lack of background information, the reader does not know whether to treat the result that there is no correlation of SH mean and IQR to the phenomenon of debris flow generation as a surprising result (because it contradicts a body of published knowledge), or instead as a demonstration that the hypothesis was negated here and may likely also be incorrect in other locations. Âż

[Reply] Measurement stations of Schmidt hammer values were selected considering a statistic analysis of lithology, structural context, and geomorphology of the studied area. Details of this work are in the undergraduate thesis of Fredes (2016) (http://repositorio.uchile.cl/handle/2250/140357) (now in reference). In this analysis we take into account a background-review the geology 1:100.000 of Sernageomin (Salazar et al.2013) and geomorphological studies of two Ph.D. thesis of the Universite de Toulouse and Universidad Catolica del Norte (Aguilar, 2010; Cabré, 2019). We include in the corrected manuscript a section "2. Study area" with a background-review of the geology and geomorphology of the Huasco river valley. We will cite in the method section works that benefit from Schmidt hammer measurements to quantify resistance to rock weathering in catchments. Nevertheless, the validation of the Schmidt Hammer is far from being the focus of this paper.

2. Methods

P3 line 18-19: « In these cases, we estimated 1 meter of debris flow thickness on average for each fan based on mean field observations.»

Comment R2: « this statement is clear. But it is not clear what it implies: Is the remaining volume of each fan treated as alluvial sediment that is NOT debris flow material? Or is it treated as an older stage of debris flow material? »

Change: « Based on the fieldwork measurements, a thickness of one meter was considered for the fans whose length and width were measured on RapidEye images.»

3. Results

P4 line 22-23: «In which amongst all, erosion of the upper mantled-hillslopes layer occurred when water concentrated and formed rills or gullies (Fig. 2 and Fig. 3).»

Comment R2: « This phrase is unclear.»

Change: «The most widespread indicators of hillslopes erosion are rills and gullies generated when water was concentrated and confined in streams (Fig. 2 and Fig. 3).»

P4 line 32-33: «Hillslopes or gravitational landslides and rockslides are the main sediment sources that characteristically fill these channels within storm periods and after storm events.»

Comment R2: «his sentence is not clear, and may be out of best order with respect to the paragraph. "Hillslope or gravitational" seems like a strange combination to connect with an "or", one word related to a landform and the other to a force.»

Change: « Gravitational landslides and rockslides of hillslopes the main sediment sources that characteristically fill alluvial channels within storm periods and after storm events.»

P5 line 7-8: «Debris flows that reached the tributary junctions during the March 2015 event were reported in forty-nine outlets out of one hundred twenty-four catchments (Fig. 4).»

Comment R2: «this phrase is ambiguous, and it is important. Each catchment must have an outlet, one would think. So one interpretation of this phrase is that there are 124 catchments, and 49 of those catchments exported a debris flow. However, the reference to "outlets" at one part of sentence and "catchments" in the other part of sentence raises the question of whether one is a subset of the other, or not. The label "ND" appears to signify No Data for about 68 catchments. Yet that adds to the ambiguity, because "no data" is not the same as "we have no debris flow here", because the absence of a debris flow IS data.»

Change: « Debris flows that reached the tributary junctions with the trunk valley and produced deposit greater to 500 m3 of sediment during the March 2015 event were reported in forty-nine of one hundred twenty-four catchments (Fig. 4).The remaining seventy-five catchments did not yield debris flows deposits greater to 500 m3 of sediments in the tributary junctions.»

P6 line 1: «The inverse correlation is also observed in the percentage of catchments that generated debris flows because the percentage increases with Size Factor while it decreases with the increase of Relief Factor (Fig. 6ab).»

Comment R2: «Note big arithmetic error on table 1)»

Reply: There's no error. Since the percentage is calculated with the basins that share in range of values and not on the total of studied basins, i.e.: Size factor: 18% of catchments with size factor 0.05-0.25 (4 of 22), 39% of catchments with size factor between 0.25-0.75 (30 of 74) and 57% of catchments with size factor 0.75-1.50 (13 of 23). Relief factor: 9% of catchments with relief factor 1.75-2.00 (1 of 11), 33% of catchments with relief factor between 1.75-1.25 (15 of 46) and 50% of catchments with relief factor 1.25-0.25 (31 of 62). We include the number of catchments in the table and change the phrase to: «The inverse correlation is also observed in the percentage of catchments with different range of factors, because the percentage of catchments that generated debris flows increases with Size Factor while it decreases with Relief Factor (Fig. 6ab).»

P6 line 12-13: «Finally, the weighting factor calculated by the PCA resulted in a normalized catchments-clustering is added (Fig. 7).»

Comment R2: «phrase is unclear. Needs a verb somewhere.)»

Change: «Finally, the weighting factor calculated by the PCA resulted in a normalized catchments-clustering. This catchments-clustering is added in a geographic information systems and resultant in a map of susceptibility (Fig. 7).»

4. Discussion

P7 line 23-24: «Recent studies of debris flow generation assessment show that soil moisture and shallow debris-mantled hillslopes failures, during intense and low frequency storm events, are not required to trigger debris flows in arid catchments (Vergara et al., 2018). This is favored by the catchment transport-limited conditions charac-
teristic of arid catchments, where debris entrainment by run-off from alluviated chan-
nels can occur from any storm that affects the area (Coe et al., 2008; Kean et al.,
2013).»

Comment R2: The next several sentences would be greatly clarified if this phrase
specifies whether you are referring to debris flows within channels or debris flows on
hillslopes, or both, or neither. Comment R2: this sentence is constructed in a confusing
manner. It appears to say, 1st, soil moisture is not required to trigger debris flows
(even during a rain event the soil remains dry?). 2nd, it appears to say that shallow
hillslope failures are not required (I have no idea how "debris-mantled" fits into the
failure statement). Is this a correct understanding of the sentence?

Change: «Recent studies of debris flow generation assessment show that shallow
debris-mantled hillslope failures is not required to trigger debris flows in arid catch-
ments during intense and low frequency storm events (Vergara et al., 2018). Hillslope
stability is favored by the transport-limited conditions characteristic of arid catchments.
Furthermore, transport limited condition favor the storage of sediment in the alluvial
channels, where debris entrainment to tributary junction alluvial fans by run-off from
alluvial channels can occur at any storm that affects the area (Coe et al., 2008; Kean
et al., 2013).»

P8 line 21-23: «So, the high altitude of zero-isotherm during the March 2015 storm
explains the great debris flow generation in the studied zone because the area with
effective water capture, as well as the distribution and the magnitude of water discharge
down system, is great.»

Comment R2: «the content is appropriate, but the sentence is somewhat unclear.»

Change: «So, the high altitude of zero-isotherm during the March 2015 storm explains
the great debris flow generation in the studied zone. In fact, greater area with effective
water capture resultant in an widespread distribution of run-off in the head-watershed

and higher volumes of water discharge downstream.»

P8 line 26-28: «In this context, the selective activation of tributary catchments and debris supply from channels by run-off during the March 2015 storm depends on the heterogeneous distribution of storm cells and on the hydrological conditioning factors to store sediments during periods without storms.»

Comment R2: «This sentence needs to clarify that the available data leads the authors to hypothesize the two controls stated, even though this study lacks the data most suitable data with which to test the hypotheses.»

Reply: Phrase deleted because the subject is included in the next paragraph.

P9 line 14-18: «This has been evidenced in the Holocene alluvial fan stratigraphy by a number of cohesive debris flow layers interpreted as a result of episodic high-water discharge events registered in the fans of El Huasco river valley (Cabré et al., 2019). Alluvial fans present at the tributary junctions; the highest sediment yield, in volume, during the relatively arid periods in the Mid-Holocene (Cabré et al., 2017, 2019). Therefore, stormy conditions and high sediment discharge at least occurred after 8 ka BP.»

Comment R2: «this is not a sentence in structure, and I cannot understand its message. How are the "number of cohesive debris flow layers" related to the "highest sediment yield"?»

Comment R2: «unclear meaning. Did Cabre et al 2019 provide chronological information which shows that the alluvial fan deposits of interest span the time from 8 to 0 ka? I ask because the previous mentions of age in this paragraph refer only to 8-4 ka, and to Mid-Holocene, not to late Holocene.»

Change: «This has been evidenced in the Holocene stratigraphy of the alluvial fans of El Huasco river valley by a number of cohesive debris flow layers and radiocarbon age, interpreted to result from episodic high-water discharge events during the last 8

ka BP (Cabré et al., 2017). Therefore, stormy conditions and high sediment discharge at least occurred after 8 ka BP.»

P9 line 25-26: «Wherever, a return of 118 years for the storms like March 2015 can be proposed for the southern Atacama Desert during the last 5,500 years (Ortega et al., 2019).»

Comment R2: «Unclear where this 118 year return value comes from, since the previous part of this paragraph tells us that Ortega et al. 2019 reported a return time of 1 event/40 years to 1 event per 210 years. These numbers seem to have nothing to do with a 118 year return time.»

Reply: 118 years take account the average during the last 5,500 years, e.i. 1 event/40 years during the last 1,000 years result in 25 events and 1 event/200 years between 1,000 to 5,500 years BP result in 22,5 events. 47,5 event in 5.500 years result in 116 years as average. We change to: «A return of 116 years for the storms like March 2015 can be proposed as average for the southern Atacama Desert during the last 5,500 years.»

P9 line 34- P10 line 1-2: «The similarity with the long-term erosion rates suggests that erosion rates have not decreased during the last 8 Ma and that very slow erosion results in an uncoupled landscape stablished at least since the Miocene Andes uplift (Aguilar et al., 2011).»

Comment R2: «Similarity of what? If you are referring to the rate inferred for the March 2015 storm, state the rate and the reference to March 2015. The rate given earlier is 1.3 mm/event. Until you integrate this over time (which occurs in a following paragraph), the reader cannot understand the comparison.»

Comment R2: «unclear what is meant by "uncoupled". If this is important, then material in the introductory section would be needed to prepare the reader for this discussion.»

Reply: We refer to the similarity between the erosion rates on a scale of thousands of

years (Aguilar et al., 2014) and the erosion rates calculated by Aguilar et al., 2011 for the last 8 million years (Aguilar et al., 2011). We change to: ««The similarity between erosion rates during the last thousands of years (Aguilar et al., 2014) with those calculated during the last 8 Ma (Aguilar et al., 2011, suggests that long-term erosion rates remain unchanged.»

P10 line 8-9: «Therefore, these two independent proxies of long-term denudation show a great significance of erosion linked to extreme storms like the March 2015 storm.»

Comment R2: «meaning is unclear. The sentence needs to be rewritten.»

Change: «These two independent proxies of long-term denudation show a great significance of erosion linked to extreme storms at scale of 106-104 years.»

Figure 1: « (a) Synoptic maps of daily precipitation during the March 23–26th, 2015 storm in the northern region of Chile (data from TRMM 3B42v7 mission). (b) Topography extracted from a Digital Elevation Model....»

Comment R2: «I don't see much value to the TRMM data, in the context of this paper. And it has been shown that the TRMM approach worked poorly for these desert region rain events.»

Reply: The figure 1a was eliminated.

Fig. 3: «Before and after from optical imagery retrieved from Planet Team (2017) showing gullies evidences after March 2015 storm. Arrows indicate different evidences of erosion processes.»

Comment R2: «caption should be more informative. At the least, it should be stated that left sides are "before" and right sides are "after". We also need to know whether the general color tone change is a physical evidence of erosion due to the March event, or if it merely indicates different sun illumination.»

Change: «Optical imagery retrieved from Planet Team (2017) before and after March

2015 storm from . Left images are before the March 2015 storm and right images are after. Arrows indicates different physical evidence of erosion processes due to the event and gully presence after March 2015 storm. The general color tone change merely indicates different sun illumination.»

Please also note the supplement to this comment:
https://www.nat-hazards-earth-syst-sci-discuss.net/nhess-2019-239/nhess-2019-239-AC5-supplement.pdf

───────────────────────────────

**Supplement:**

[revised manuscript text omitted]
 within the catchments, and these are significant to Hydrometeorological hazards assessment.

**2 Study area**

El Huasco river valley is one of the main fluvial systems in the climatic transition area between the hyper-arid core of the Atacama Desert and the semiarid region of the Central Chile. This watershed has two main rivers: El Carmen River and El Tránsito River. These two rivers have permanent runoff from snow and permafrost melting from the higher tributaries. Currently glacial and periglacial environment are present above 4000 m a.s.l. and Late Pleistocene glacial evidences are not present below 3200 m a.s.l. (Aguilar, 2010).

The mid-mountainous zone of this fluvial system situated between 3500-1000 m a.s.l. (Fig. 1), is usually impacted by spatially heterogeneous extreme storm events which trigger shallow landslides and debris flows in the tributary catchments. The episodic sedimentation associated to individual storm events is recorded in the Holocene stratigraphic arrangements in some of the tributary-junction alluvial fans (Veit, 1996; Aguilar, 2010; Cabré et al., 2017, 2019). Authors identified a number of cohesive debris flows in different alluvial fans indicating the episodic nature of sedimentation whereas gravel-sheet couplets and open framework gravels indicate more continuous sedimentation in the alluvial fans associated to relatively more humid periods within the Holocene (Cabré et al., 2019).

The quantification and characterization of erosion and sedimentation in the Huasco river valley has been interpreted by different authors as a climatic and tectonic induced signal (Aguilar et al., 2011, 2013, 2014; Rossel et al., 2018). At a $10^6$-year time-scale, tectonic and climatic factors which combines the aridization and the Andean uplift are coupled to develop a transient landscape organization (Aguilar et al., 2013). The calculated mean erosion rates of 0.03-0.08 mm/yr during the last 10-6 Ma not reach the fully degradation of the Miocene surfaces situated at the highest altitudes (Aguilar et al., 2011). Millennial erosion rates ($10^4$-year) measured by means of Terrestrial Cosmogenic Nuclides concentrations in the river sediments agree with $10^6$-year time-scale erosion rates (Aguilar et al., 2014).

Permo-Triassic igneous-metamorphic and Mesozoic rocks are exhumed during the Cenozoic and represent the highest mountain ranges in the area. The bedrock is dominated by Permo-Triassic igneous-metamorphic rocks which are subdivided based on Salazar et al. (2013) into: Pampa Gneiss, Tránsito Metamorphic Complex and Chanchoquín Plutonic Complex. Mesozoic Volcano-sedimentary units are also present between these Permo-Triassic rock-blocks: San Félix Fm., Lautaro Fm., and Lagunillas Fm. These lithological units have been intruded by several Cenozoic plutons and are overlain by Cenozoic unconsolidated gravel deposits (e.g. Atacama Gravels) and volcanic deposits (Salazar et al., 2013).

Five geological groups were defined by Fredes (2016) considering a rock strength classification (González de Vallejo, 2002) of the geological units of Salazar et al. (2013) (Fig. 9a). Geo1 corresponds to intrusive rocks, Geo2 corresponds to volcanic and coarse-sedimentary rocks, Geo3 corresponds to metamorphic rocks, Geo4 corresponds to fine grained rocks (shales and siltstones) and Geo5 corresponds to unconsolidated Pleistocene-Holocene deposits. This strategy and the measurement of Schmidt hammer values has been applied in the Morocco Atlas by Stokes and Mather (2015) to characterize the lithological influence in the debris flows generation.

**3 Methods**

Immediately after the March 2015 storm we performed a field survey in El Huasco river valley. We observed (1) debris flow deposition on the alluvial fan surfaces and (2) alluvial fan formation in the confluence of the tributary with the trunk valley. Analysis of deposits showed different rheologies of Debris flows ranging from cohesive debris flows and hyper-concentrated flows to mud flows. Here, to simplify the statistical analysis we do not differentiate between flow-types and henceforth we will include all under the "debris flow" term sensu stricto. For details on the spatial and temporal distribution of debris flows pulses in the fans, readers are referred to Cabré et al. (submitted).

We have calculated the mean erosion (mean landsurface lowering in millimeters) after the March 2015 storm event for each of the tributary catchments and for the whole catchments within the study area with the debris flow deposits volumes (Fig. 1, tributaries inside dashed-line white box). Tributary catchment were extracted use of a Geographic Information System. The Digital Elevation Model is provided by 
[revised manuscript text omitted]
 this study in agreement with previous reports of arid zone hillslope responses to rainfall that have quantified the contribution from gullies to be 50 to 80% of the overall sediment yield (Poesen et al., 2003).

The selective generation of debris flows within tributary catchments has been reported previously in Andean catchments (Colombo, 2010; Lauro et al., 2017). This selective activation might be explained by different coupling degrees between catchments and trunk valley (Fryirs et al., 2007; Mather and Stokes, 2017) or by the absence of the necessary rainfall amount to trigger debris flows in some catchments linked to the heterogeneous rainfall distribution of ENSO events (Colombo, 2010; Cabré et al., 2019). In that sense, the selective activation of a few catchments for El Elqui river valley was reported 130 km south of the study area (Vergara et al., 2018). Vergara et al. (2018) also highlights that the intensity of the 1-h peak storm precipitation had a low significance in the distribution of debris flow generation during the storms. The results of the logistic regression model in El Elqui river valley (Vergara et al., 2018) supports that the prediction of high discharge events does not depend on the antecedent precipitation (accumulated rainfall).

The entrainment of sediments from the drainage network during the March 2015 storm agrees with the topographical attributes of catchments that are used to identify significant debris flow generation. We have shown that the topographical attributes play a major role in the debris flow generation and distribution after extreme storms. Accordingly, most of the largest tributary catchments, with highly developed stream-networks and relatively low mean slopes, were activated during the March 2015 event. The large volumes of sediments stored in gullies and channels resulted in several debris flows during the storm that yielded large volumes of sediment to the outlets (Fig. 10). In contrast, small and steep catchments, with low developed drainage network, store less volumes of sediments in the channels and thus during extreme storm events generate flows with lower sediment concentrations.

The altitude of zero-isotherm in this region of the Andes appears to greatly influence debris flow generation during high discharge events related to extreme precipitation events (Moreiras et al., 2018; Vergara et al., 2018). So, the high altitude of zero-isotherm during the March 2015 storm explains the abundant debris flow generation in the studied zone. In fact, greater area with effective water capture resultant in an widespread distribution of run-off in the head-watershed and higher volumes of water discharge downstream. However, altitudinal attributes of catchments are not identified as conditioning factors of the selective distributions of debris flow generation during the March 2015 event in the studied zone, because during this storm the zero-isotherm is above the maximum altitude of the topography.

Susceptibility assessment to debris flow generation must be evaluated in hydro-meteorological hazard studies for populated area (Wilford et al., 2004). In one sense, susceptibility of debris flow generation in the southern Atacama Desert is linked to the capacity of sediment-storage in a drainage network during the inter-storm period. Beside, susceptibility is linked to the percentage of the catchment over the altitude of zero-isotherm during the storm. The unusual altitude of zero-isotherm above the topography during the March 2015 storm explains the lack of similar sedimentary-behavior in the historical register and the extensive runoff of debris flows from the source of sediments. Considering debris flow susceptibility studies for a storm with a high zero-isotherm is relevant when considering its progressive increase of typical elevation of the zero-isotherm in Chile (Carrasco et al., 2005; Boiser et al., 2016) and in mountain areas worldwide (Mountain Research Initiative EDW Working Group, 2015).

**5.2 The role of individual extreme storm events in erosion rates in the southern Atacama Desert**

The occasional humid spells reported for the Mid-Holocene (8-4 ka) (Grosjean et al, 1997) or stormy conditions (Tiner et al., 2018) produced the necessary runoff to entrain sediment from the alluvial channels and hillslopes within tributary catchments of the Atacama Desert. This has been evidenced in the Holocene stratigraphy of the alluvial fans of El Huasco river valley by a number of cohesive debris flow layers and radiocarbon age, interpreted to result from episodic high-water discharge events during the last 8 ka BP (Cabré et al., 2017). Therefore, stormy conditions and high sediment discharge at least occurred after 8 ka BP.

The increase of layers with coarse sediment in both lacustrine and marine archives during the last 5,500 years BP in Northern Chile are interpreted as high sediment discharge events associated with more recurrent extreme storms in the southern Atacama Desert (Rein et al., 2005; Tiner et al., 2018; Ortega et al., 2019). The recurrence time decreased from 1 event/210 yr towards 1 event/ 40 yr in the last 1,000 years BP (Ortega et al., 2019). Nevertheless, the increase of coarse-sediment layers could be showing more summer-austral storms, with high zero-isotherm altitude like the March 2015 storm. Therefore, this sedimentary record does not necessarily show changes of inter-decennial storm-return time linked to ENSO as interpreted by Ortega et al. (2019). A return of 116 years for the storms like March 2015 can be proposed as average for the southern Atacama Desert during the last 5,500 years based on Ortega et al. (2019). The return time proposed for the southern Atacama Desert is similar to the 100 years return time estimated by Houston (2006) in El Salado River a tributary of El Loa Valley in the Altiplano, where the influence of storms related to the warm phase of ENSO is low.

The role of extreme storm events in erosion rates should be considered only for data that encompass extensive areas or an exhaustive correlation between specific places because focused assessment might not be representative of a great magnitude storm like the March 2015 event. Quaternary erosion rates from concentration of Terrestrial Cosmogenic Nuclides in stream sediments of El Huasco river valley are reported in Aguilar et al. (2014). Their results presented denudation rates of 0.03-0.05 mm/yr and of 0.06-0.08 mm/yr, measured in sand and gravel grain-size fractions, respectively. The time period of denudation rates calculated by Aguilar et al. (2014) is between 20 ka for sand and 12 ka for gravels. The similarity between erosion rates during the last thousands of years (Aguilar et al., 2014) with those calculated during the last 8 Ma (Aguilar et al., 2011), suggests that long-term erosion rates remain unchanged (
[revised manuscript text omitted]

---

## Author Comment (AC6) · 18 Nov 2019

Dears reviewer and editor, following I include specific responses to your comments and corrections. Attached pdf file is the corrected manuscript include all modifications. Thank you very much for your great contribution in ours manuscript.

Best Wishes

German Aguilar

Specific Answers to Reviewer 3

Line 3-4, p6: question: This statement refers to debris flows reaching the main valley, isn't? I mean, it probably there were debris flows within the catchment but no big enough to deliver sediment to the outlet alluvial fan? [Reply] The percentage of catchments that generated debris flows were calculated considering if the flows reached the trunk rivers. It is very probable that was generated debris flows in other catchments and that did not reach the trunk rivers. We will clarify this in the corrected manuscript: « .... debris flows were generated in the tributary junctions....».

Line 14, p6: regarding positive correlation you mention: the higher the relief factor the steeper the slope within the catchment? therefore negative correlation with volumes of debris flows? [Reply] Indeed, there is an error that we solve in the corrected version: The relationship is negative between the volume of sediments and the relief factor.

Please also note the supplement to this comment:
https://www.nat-hazards-earth-syst-sci-discuss.net/nhess-2019-239/nhess-2019-239-AC6-supplement.pdf
* * *

---

## Author Response (AR2)

Dear Editor,

On behalf of my co-authors, I would like to submit the corrected manuscript entitled "Erosion after an extreme storm event in an arid fluvial system of the southern Atacama Desert: an assessment of magnitude, return time, and conditioning factors of erosion and debris flows generation". We feel that our corrected version after Interactive comments in NHESSD is a valuable contribution to the understanding of the extreme erosion processes in the Atacama Desert. I thank you for the consideration of this work for publication in the journal Natural Hazards and Earth System Sciences (NHESS).

Sincerely,

German Aguilar

Specific responses to the reviewer comments

Page 3 line 28: it is quite strange that the first cited figure in the manuscript is figure 9. Please, consider revising the order of figures.

Reply 1: This is a mistake after a change in the manuscript organization of an ancient version. Now is corrected with the change in figure order. Fig 9a => Fig. 2 and Fig 9b => Fig 10
* * *
Page 3 line 30: it is not clear what "this strategy" means; could be referred to the Terrestrial Cosmogenic Nuclides concentrations measurement but it is not clear, I suggest revising the organisation of this part of the chapter.

Reply 2: We reviewed the content of the sentence: "This strategy and the measurement of Schmidt hammer values has been applied in the Morocco Atlas by Stokes and Mather (2015) to characterize the lithological influence in the debris flows generation". We eliminated this sentence because it is out of context (remain after manuscript re-organization of an ancient version). The scope of this sentence is well located at the end of the methods chapter.
* * *
Page 4 line 7: I am confident that the submitted paper Cabré et al. (submitted) will be published, but I suggest improving the description of the debris flow deposit to make this manuscript more detailed.

Reply 3: Indeed, Cabre et al. (2020) have already published (https://doi.org/10.1177/0309133319898994). However, we include a table (Table 1) summarizing the main characteristics of sedimentary facies of debris flow deposits.
* * *
Chapter 2 general comment: the description of the study area is too limited. Readers are not able to know the extension of the area, the name of basins, and other necessary geomorphological information. This chapter should be thoroughly revised and partially rewritten.

Reply 4: We improved the first paragraph of this chapter to provide more physiographic and geomorphological information. We include: area of the study zone, altitudes and other general

characteristics. Figure 1 was modified to show these informations.The detailed topographic, morphometric and geological information of the 124 tributary catchments studied are in appendix. Also, this chapter was fully revised and partially rewritten.
* * *
Figure 1: this picture is presented a lot of information not cited in the text. I do not think that this is correct. Please, consider the possibility to add a chapter with the description of available data.

Reply 5: The information of the Figure 1 now are partially cited in the text of the chapter 2 and mainly in the new chapter 3 titled: Rainfall data and erosion processes operating during the March 2015 storm event.
* * *
Chapter 3: this is the most important chapter of the manuscript, and it presents a lot of problems and limitations. First of all, readers are not able to understand exactly what the authors did. There a sequence of general information about methods and approach, but is not easy to understand what exactly was performed in this area. Did authors use the formula (1) or the ANOVA approach, or both? I suggest considering a workflow that describes better what authors exactly did.

Reply 6: We separate the chapter into sub-chapters for better reading and understanding of the methodology workflow involved.
* * *
Chapter 4.1: this is a general description that has already done by authors in the previous part of the manuscript.

Reply 7: We considered this observation and now the information provided by sub-chapter 4.1 has been moved to the new chapter 3, taking care not to repeat the information. So chapter 4.1 was deleted.
* * *
Chapter 4.2: This is an important point of the manuscript. Volume estimations of debris flows are provided but in the chapter "method" there is not a real description of the accuracy of the presented method. Even in this chapter, volume estimations are presented without an explanation of the accuracy of the method and a definition of the tolerance of the evaluations. The definition of volumes expressed in meters without a range that describes the tolerance of the estimations is a big limitation of this paper.

Page 6 line 16: The sentence: "The propagation of errors entails an uncertainty of 10% for the volumes calculated with field measurements and 40% for volumes estimated by satellite image measurements" cannot be considered sufficient for the definition of uncertainty. A better description of the procedure adopted for the definition of these values should be provided by authors.

Reply 8: Now results chapter is the 5. We include a table as supplementary data with all the volume calculations made and the accuracy of the method. Also in the new version the uncertainty of the data is better showed and supplementary data are cited in the main text..
* * *
Page 7 line 5: the lack of description of the studied area has a strong impact in this chapter. Readers are not able to understand which (and how many) catchments produced debris flow and which are characterised by clean water flow.

Page 7 line 9: "Debris flows occurred only in 9% of the catchments" how many catchments are considered in this papers? I do not find a detailed description of this basic information.

Reply 9: Now results chapter is the 5. The first sentence of subchapter 5.1 says in how many catchments debris-flow deposits were generated (49) and in which they did not develop (75). This information is also clearly expressed in figures and tables. However, at the request of the reviewer, this information was given again in the paragraph mentioned in sub-chapter 5.2 and 5.3. Also in the chapter of methodology (4) we include the total of studied tributary catchments (124). I would like to clarify that the inventory of catchments where debris flows were deposited is part of the results of this work and does not represent the framework of the study area. Note also the improve of the description of the studied area (Chapter 2 and 3).
* * *
Chapter 4.4: this chapter is very short, and the description of obtained results it is not very clear.

Reply 10: We're not quite sure what the reviewer is referring to at this point. The text is short, nevertheless the related figures use a lot of data and its cleary presented. We think that no modifications are necessary at this subject.
* * *
Page 8 line 19. "We propose based on field observations that this is because debris supply from rocky outcrops of hillslopes is not a significant source of sediments during the storms. Instead, we propose that the main source of debris flows is sediment stored in the drainage network." This is a good point, but I am not sure that this comment is clearly supported by presented data.

Reply 1: Now we rewrite the paragraph.
* * *
Page 9: this page should be totally rewritten. It is not clear the real objective of this part of the manuscript. There is a critical mix between data of the studied basin and data of other basins studied by other authors and not related to the 2015 event. This data, in my opinion, should be presented earlier and the authors can make a discussion pointing out differences between the studied phenomenon and other case studies.

Reply 12. Paragraphs in this page, and in the whole chapter of discussion, were rewritten, considering the comments of the reviewer. Now, we present first the interpretation of our results, and later include the discussion of its significance in relation with other works.
* * *
Page 10 line 1: the mean value of 1.3 mm of erosion has been calculated without an effective description of the accuracy of the method. The lack of a serious scientific method is very critical.

[Figure]

Reply 13: see Reply 8.
* * *
Page 10 line 3: "and the return time of these kind of events from paleoclimatic record of 1 event each 100 years." I did not find in the manuscript this paleoclimatic study before this sentence.

Reply 14: The reviewer refers to line 3 of page 11, but in the same chapter we refers to paleo-climatic record of others works. We think that no modifications are necessary at this subject.
* * *
Page 11 line 10 "The limitations to study erosion associated with an individual storm event include the lack of pre-event high resolution topography" in this paper, I do think that authors mentioned the availability of a detailed topography after the event.

Reply 15: Indeed, there is a detailed topography (LIDAR) taken after the event, but not before the event and not for the whole study area. We think that no modifications are necessary at this subject.
* * *
Page 11 line 15: "Nevertheless, these records also underestimate the peak discharge associated with the storm because the gauging stations are commonly damaged by flash floods" Is this e general comment or referred to the presented case study?

Reply 16: This is a general statement, at least for the Huasco River and other nears rivers of Chile. For example, in this year, the governmental of Chile inaugurate a more robust station in the Huasco River (https://atacamanoticias.cl/2019/11/14/nueva-estacion-fluviometrica-el-maiten-en-el-rio-huasco-lleva-un-85-de-avance/). We think that no modifications are necessary at this subject.

---

## Author Response (AR3)

Dears,

I am pleased to submit our accepted paper "Erosion after an extreme storm event in an arid fluvial system of the southern Atacama Desert: an assessment of magnitude, return time, and conditioning factors of erosion and debris flows generation" for publication in the journal Natural Hazards and Earth System Sciences (NHESS). Text, figures and supplementary data are included according to the required formats.

Sincerely yours,

German Aguilar